# Noisy SIGNSGD Is More Differentially Private Than You (Might) Think

**Richeng Jin** [1]  **Huaiyu Dai** [2]

## Abstract

The prevalent distributed machine learning paradigm faces two critical challenges: communication efficiency and data privacy. SIGNSGD provides a simple-to-implement approach with improved communication efficiency by requiring workers to share only the signs of the gradients. However, it fails to converge in the presence of data heterogeneity, and a simple fix is to add Gaussian noise before taking the signs, which leads to the Noisy SIGNSGD algorithm that enjoys competitive performance while significantly reducing the communication overhead. Existing results suggest that Noisy SIGNSGD with additive Gaussian noise has the same privacy guarantee as classic DP-SGD due to the post-processing property of differential privacy, and logistic noise may be a good alternative to Gaussian noise when combined with the sign-based compressor. Nonetheless, discarding the magnitudes in Noisy SIGNSGD leads to information loss, which may intuitively amplify privacy. In this paper, we make this intuition rigorous and quantify the privacy amplification effect of the sign-based compressor. Particularly, we analytically show that Gaussian noise leads to a smaller estimation error than logistic noise when combined with the sign-based compressor and may be more suitable for distributed learning with heterogeneous data. Then, we further establish the convergence of Noisy SIGNSGD. Finally, extensive experiments are conducted to validate the theoretical results.

## 1. Introduction

Nowadays, deep learning has been playing a vital role in various applications, which is fundamentally reshaping the development of our modern society. With the tremendous growth of neural network size, it is unlikely, if not impossible, to store all the training data on a single machine and perform centralized training. Therefore, distributed learning has become the most prevalent training paradigm. In the typical parameter server framework, a set of workers (or equivalently clients) collaboratively train a global model under the coordination of a server. In the celebrated distributed stochastic gradient descent (SGD) algorithm (Bertsekas & Tsitsiklis, 2015), the server maintains a model that is updated iteratively by aggregating the stochastic gradients derived on the local datasets from the workers.

Distributed SGD faces two critical challenges. As the sizes of neural networks grow, the communication overhead incurred for the transmission of gradients becomes the major bottleneck. For instance, modern large language models may have billions of model parameters, and sharing the gradients leads to prohibitive communication overhead. In the meantime, in applications like federated learning, the local training data may contain sensitive information about the workers, which renders them unwilling to participate in training. Particularly, it has been shown that training data can be effectively reconstructed from the gradients (Wang et al., 2019; Zhu et al., 2019; Yue et al., 2023), resulting in privacy concerns.

To address communication efficiency concern, SIGNSGD, which allows the workers to transmit the signs of gradients and enables an improvement of $32\times$ in communication efficiency, is proposed (Bernstein et al., 2018; 2019). One major concern that prevents the application of the vanilla SIGNSGD is the non-convergence issue in the presence of data heterogeneity across workers. Therefore, various methods, including error compensation (Karimireddy et al., 2019; Zheng et al., 2019), stochastic sign compressor (Chen et al., 2020b; Safaryan & Richtárik, 2021; Jin et al., 2024; Tang et al., 2024), variance reduction (Chzhen & Schechtman, 2023), local momentum (Sun et al., 2023), and adaptive methods (Crawshaw et al., 2022), have been proposed to address the non-convergence issue.

Among these approaches, a simple and elegant fix is adding

[1]College of Information Science and Electronic Engineering, Zhejiang University, Hangzhou, China. [2]Department of Electrical and Computer Engineering, North Carolina State University, Raleigh, NC, USA.. Correspondence to: Richeng Jin <richengjin@zju.edu.cn>.

*Proceedings of the $42^{nd}$ International Conference on Machine Learning*, Vancouver, Canada. PMLR 267, 2025. Copyright 2025 by the author(s).

Gaussian noise to the gradients before taking the signs, which leads to the G-NoisySign compressor and the corresponding Noisy sIGNSGD algorithm (Chen et al., 2020b). Meanwhile, adding Gaussian noise is the de facto method for preserving differential privacy (DP) in distributed learning, which has been shown to be order-optimal (Cai et al., 2021). Thanks to the post-processing property of differential privacy, the privacy guarantee is preserved after taking the signs of the perturbed gradients (Jin et al., 2020). (Chaudhuri et al., 2022) and (Guo et al., 2023) empirically observe that Noisy sIGNSGD achieves a comparable performance to the classic Gaussian mechanism without compression. Recently, (Jang et al., 2024) shows that logistic noise (with the corresponding compressor denoted by L-NoisySign) is more appropriate than Gaussian noise for sign-based gradient descent methods. However, these works do not account for the potential privacy amplification effect of $sign(\cdot)$. Intuitively, discarding the magnitudes of the noisy gradients further incurs information loss, which may potentially improve privacy preservation. In this work, we make this intuition rigorous by theoretically deriving the differential privacy guarantee of the generic NoisySign compressors and investigating their performance in distributed learning. In particular, this paper makes the following contributions.

- We theoretically analyze the differential privacy guarantee of the generic NoisySign compressor that captures G-NoisySign and L-NoisySign as special cases and show the privacy amplification effect of the $sign(\cdot)$ compressor through the lens of $f$-DP. Particularly, for the scalar case, the derived privacy guarantee is tight, in the sense that it cannot be improved in general.

- Based on the theoretical results, we further show that given the same privacy guarantee, G-NoisySign enjoys a smaller estimation error than L-NoisySign and may be more suitable for distributed learning scenarios with heterogeneous data.

- We establish the convergence of Noisy sIGNSGD with two aggregation schemes: averaging and majority voting. Extensive numerical and experimental results validate our theoretical findings.

## 2. Related Work

**Sign-based Gradient Quantization:** The idea of sharing the signs of gradients in SGD can be traced back to 1-bit SGD (Seide et al., 2014). Despite that sign-based quantization is biased in nature, (Carlson et al., 2015) and (Bernstein et al., 2018; 2019) show theoretical and empirical evidence that sign-based gradient descent schemes converge well in the homogeneous data distribution scenario. (Safaryan & Richtárik, 2021) shows the convergence of sIGNSGD given the assumption that the probability of wrong aggregation is less than $1/2$. In the heterogeneous data distribution case,

(Chen et al., 2020b) shows that the convergence of sIGNSGD is not guaranteed and proposes to add carefully designed noise to ensure a convergence rate of $O(d^{\frac{3}{4}}/T^{\frac{1}{4}})$. (Jin et al., 2024) proposes a stochastic-sign compressor to address the non-convergence issue. However, none of these works takes privacy into consideration.

**Differential Privacy Mechanism**: To improve communication efficiency over the classic DP-SGD (Abadi et al., 2016), significant research efforts have been devoted to developing discrete mechanisms. (Agarwal et al., 2018) extends the one-dimensional binomial noise scheme (Dwork et al., 2006a) to the general $d$-dimensional case with more comprehensive analysis in terms of $(\epsilon, \delta)$-DP. (Canonne et al., 2020; Kairouz et al., 2021) study the DP guarantees of discrete Gaussian noise. (Agarwal et al., 2021) and (Chen et al., 2022) propose the Skellam mechanism and the Poisson binomial mechanism, respectively, with Rényi DP guarantees. (Chaudhuri et al., 2022; Guo et al., 2023) propose privacy-aware compression through numerical mechanism design, and (Zhu & Blaschko, 2023) studies the impact of random sparsification on DP-SGD. (Chen et al., 2020a) proposes the subsampled and quantized Kashin's response (SQKR) mechanism that achieves order-optimal estimation error in distributed mean estimation. (Chen et al., 2023) studies privacy amplification by compression for central $(\epsilon, \delta)$-DP, while (Jin et al., 2023) considers the privacy amplification of random sparsification through the lens of $f$-DP.

The most related work to this paper is (Jang et al., 2024), which finds that logistic noise is more appropriate than Gaussian noise when combined with the sign-based compressor. It is shown that, given the same privacy guarantee, L-NoisySign has a lower sign-flipping probability than G-NoisySign. However, the sign-flipping probability may not be the best performance indicator considering that the vanilla sIGNSGD has a sign-flipping probability of 0 but fails to converge with heterogeneous data. In sharp contrast, by accounting for the privacy amplification effect of the $sign(\cdot)$ compressor, we show that G-NoisySign enjoys a smaller estimation error given the same privacy guarantee, which leads to a smaller probability of wrong aggregation in the distributed sign estimation scenario with data heterogeneity. In addition, we adopt the emerging $f$-DP framework (Dong et al., 2021), which enjoys a better composition property than the classic $(\epsilon, \delta)$-DP considered in (Jang et al., 2024).

**Compression of Differential Privacy Mechanism**: Another line of research considers the compression of differentially private mechanisms (Bassily & Smith, 2015; Triastcyn et al., 2021; Feldman & Talwar, 2021; Shah et al., 2022; Liu et al., 2024), which aims to compress and simulate the distribution of the DP optimizer, usually in the presence of some shared randomness. The resulting compressed mechanisms have a smaller communication cost compared to

the original mechanism while retaining the (or weakened) privacy guarantee. In this work, we investigate the privacy amplification of the sign-based compressor.

## 3. Problem Setup and Preliminaries

### 3.1. Problem Setup

We consider the classical parameter-server distributed learning framework, comprising $M$ workers (denoted by $\mathcal{H}$) and a central server. Each worker is equipped with a local dataset $\mathcal{D}_m$, and the collective objective is to minimize the finite-sum function:

$$\min_{\boldsymbol{w} \in \mathbb{R}^d} F(\boldsymbol{w}) \overset{\text{def}}{=} \frac{1}{M} \sum_{m \in \mathcal{H}} f_m(\boldsymbol{w}), \qquad (1)$$

where $f_m(\boldsymbol{w})$ represents the local loss function associated with worker $m$, defined by $\mathcal{D}_m$ and the parameter vector $\boldsymbol{w} \in \mathcal{W}$. Specifically, $f_m(\boldsymbol{w}) = \frac{1}{|\mathcal{D}_m|} \sum_{s \in \mathcal{D}_m} l(\boldsymbol{w}; s)$ where $|\mathcal{D}_m|$ is the size of $\mathcal{D}_m$ and $l : \mathcal{W} \times \mathcal{D} \to \mathbb{R}$ is the loss function that quantifies the error of prediction on a data point $s \in \mathcal{D}_m$ made with $\boldsymbol{w}$.

### 3.2. Differential Privacy

Differential privacy is a rigorous mathematical framework for quantifying and ensuring privacy in data analysis. At its core, DP guarantees that the output of a mechanism remains statistically indistinguishable for two neighboring input datasets that differ in only one record. Formally, it is defined as follows.

**Definition 1** (($\epsilon, \delta$)-DP (Dwork et al., 2006a)). *A randomized mechanism $\mathcal{M}$ is ($\epsilon, \delta$)-differentially private if for all neighboring datasets $S$ and $S'$ and all $O \subset \mathcal{O}$ in the range of $\mathcal{M}$, we have*

$$P(\mathcal{M}(S) \in O) \le e^\epsilon P(\mathcal{M}(S') \in O) + \delta, \qquad (2)$$

*in which $\epsilon, \delta \ge 0$ are the parameters that characterize the level of differential privacy.*

The application of differential privacy can be largely divided into two categories: (1) Central differential privacy (CDP) (Dwork et al., 2006b) requires a trusted central server to randomize the collected data from the workers and release a private version of the aggregated results; (2) Local differential privacy (LDP) (Kasiviswanathan et al., 2011) does not assume a trusted central server. Instead, the workers perturb the data before sharing them with the central server. In this work, similar to (Jang et al., 2024), we mainly focus on LDP.

### 3.3. $f$-Differential Privacy

Consider two neighboring datasets $S$ and $S'$, from the hypothesis testing perspective, there are two hypotheses

$$\begin{aligned} H_0 &: \text{the underlying dataset is } S, \\ H_1 &: \text{the underlying dataset is } S'. \end{aligned} \qquad (3)$$

Let $P$ and $Q$ denote the probability distribution of $\mathcal{M}(S)$ and $\mathcal{M}(S')$, respectively. Following the formulation in (Dong et al., 2021), the task of distinguishing the two hypotheses can be described as a tradeoff between the achievable type I and type II error rates. Specifically, for a rejection rule $0 \le \phi \le 1$, the type I and type II error rates are defined as $\alpha_\phi = \mathbb{E}_P[\phi]$ and $\beta_\phi = 1 - \mathbb{E}_Q[\phi]$, respectively. $f$-DP characterizes the tradeoff between these two error rates by a tradeoff function formally defined as follows.

**Definition 2** (Tradeoff function). *For any two probability distributions $P$ and $Q$ on the same space, the tradeoff function $T(P, Q) : [0, 1] \to [0, 1]$ is defined as $T(P, Q)(\alpha) = \inf\{\beta_\phi : \alpha_\phi \le \alpha\}$, where the infimum is taken over all (measurable) rejection rule $\phi$.*

**Definition 3** ($f$-DP). *Let $f$ be a tradeoff function. With a slight abuse of notation, a mechanism $\mathcal{M}$ is $f$-differentially private if $T(\mathcal{M}(S), \mathcal{M}(S')) \ge f$ for all neighboring datasets $S$ and $S'$, which suggests that the attacker cannot achieve a type II error rate lower than $f(\alpha)$.*

$f$-DP can be converted to ($\epsilon, \delta$)-DP in a lossless way as follows.

**Lemma 1.** *(Dong et al., 2021) A mechanism is $f(\alpha)$-differentially private if and only if it is ($\epsilon, \delta$)-differentially private with*

$$f(\alpha) = \max\{0, 1 - \delta - e^\epsilon \alpha, e^{-\epsilon}(1 - \delta - \alpha)\}. \quad (4)$$

We further introduce a special instance of $f$-DP with $f(\alpha) = \Phi(\Phi^{-1}(1 - \alpha) - \mu)$, referred to as $\mu$-GDP. In this case, $\mu$-GDP corresponds to the tradeoff function of two normal distributions with mean 0 and $\mu$, respectively, and unit variance. It has the following composition property.

**Lemma 2.** *The $T$-fold composition of $\mu_t$-GDP mechanisms is $\sqrt{u_1^2 + u_2^2 + \cdots + u_T^2}$-GDP.*

## 4. Differential Privacy of Noisy SIGNSGD

### 4.1. DP Guarantees of Generic Sign-based Compressors

In this section, we consider a generic sign-based compressor and investigate its privacy guarantee through the lens of $f$-DP, starting with the scalar case and extending it to the vector case in Section 4.3.

**Definition 4** (**Generic Sign-based Compressor**). *For any given $x \in [-c, c]$, the generic noisy sign-based compressor outputs $NoisySign(x, p_{+1}, p_{-1})$, which is given by*

$$NoisySign(x, p_{+1}, p_{-1}) = \begin{cases} 1, & \text{with probability } p_{+1}(x), \\ -1, & \text{with probability } p_{-1}(x), \end{cases} \qquad (5)$$

*where $p_{+1}(x), p_{-1}(x) \in [p_{min}, p_{max}]$, $p_{+1}(x) + p_{-1}(x) = 1$, and $0 \le p_{min} \le p_{max} \le 1$.*

In the following, we show the $f$-DP of the sign-based compressor. We provide a sketch of proof, and the complete

proof is given in Appendix A. It is worth mentioning that the result is tight and cannot be improved in general since no relaxation is involved.

**Theorem 1.** *Suppose that* $p_{max} + p_{min} = 1$*, and* $p_{+1}(x) > p_{+1}(y), \forall x > y$*. The sign-based compressor is* $f^{NoisySign}(\alpha)$*-differentially private with*

$$
f^{NoisySign}(\alpha) = \begin{cases} 1 - \frac{p_{max}}{p_{min}}\alpha, & \text{for } \alpha \in [0, p_{min}], \\ \frac{p_{min}}{p_{max}} - \frac{p_{min}}{p_{max}}\alpha, & \text{for } \alpha \in [p_{min}, 1]. \end{cases}
$$
(6)

*Sketch of Proof.* The proof of Theorem 1 utilizes the Neyman-Pearson Lemma (Lehmann et al., 2005), which states that the most powerful test for the hypothesis testing problem (i.e., whether the input is $x$ or $x'$) at a given type I error rate $\alpha$ is threshold-based, which reduces the problem to determining the corresponding threshold. Without loss of generality, assume that $x > x'$ and let $Y = NoisySign(x', p_{+1}, p_{-1})$ and $X = NoisySign(x, p_{+1}, p_{-1})$.

We may set the threshold $h = \frac{p_{-1}(x')}{p_{-1}(x)}$ for $\alpha \in [0, p_{-1}(x)]$, i.e., if the likelihood ratio $\frac{P(Y=k)}{P(X=k)}$ is larger than/equal to/smaller than $h$, the hypothesis that the true data is $x$ will be rejected with probability $1/\gamma/0$, where $k \in \{-1, 1\}$ is the observed output of the sign-based compressor. In this case, observing that $\frac{P(Y=-1)}{P(X=-1)} > 1 > \frac{P(Y=1)}{P(X=1)}$ since $p_{+1}(x) > p_{+1}(y), \forall x > y$, we can derive the type I and type II error rates as follows.

$$
\mathbb{E}_P[\phi] = \gamma P(X = -1) = \gamma p_{-1}(x) = \alpha, \quad (7)
$$

$$
1 - \mathbb{E}_Q[\phi] = 1 - \gamma P(Y = -1) = 1 - \gamma p_{-1}(x')
$$
$$
= 1 - \frac{p_{-1}(x')}{p_{-1}(x)}\alpha. \quad (8)
$$

Similarly, when $\alpha \in [p_{-1}(x), 1]$, we set $h = \frac{P(Y=1)}{P(X=1)}$, and therefore

$$
\mathbb{E}_P[\phi] = P(X = -1) + \gamma P(X = 1)
$$
$$
= p_{-1}(x) + \gamma p_{+1}(x) = \alpha, \quad (9)
$$

and

$$
1 - \mathbb{E}_Q[\phi] = 1 - P(Y = -1) - \gamma P(Y = 1)
$$
$$
= p_{+1}(x') - \gamma p_{+1}(x')
$$
$$
= p_{+1}(x') - \frac{\alpha - p_{-1}(x)}{p_{+1}(x)}p_{+1}(x'). \quad (10)
$$

In summary, the type II error rate is given by

$$
\beta_\phi(\alpha) = \begin{cases} 1 - \frac{p_{-1}(x')}{p_{-1}(x)}\alpha, & \text{for } \alpha \in [0, p_{-1}(x)], \\ \frac{p_{+1}(x')}{p_{+1}(x)} - \frac{p_{+1}(x')}{p_{+1}(x)}\alpha, & \text{for } \alpha \in [p_{-1}(x), 1]. \end{cases}
$$
(11)

The infimum of $\beta_\phi(\alpha)$ is attained when $p_{-1}(x') = p_{max}$ and $p_{-1}(x) = p_{min}$, which completes the proof. □

### 4.2. Privacy Amplification of deterministic $sign(\cdot)$

Given the privacy guarantee of the generic sign-based compressor in Section 4.1, we examine the privacy amplification effect of the deterministic $sign(\cdot)$ compressor. Intuitively, the $sign(\cdot)$ operator does not provide differential privacy guarantees on its own due to its deterministic nature, and the adversary can distinguish $x$ from $x'$ given $sign(x)$ or $sign(x')$ if $xx' < 0$. However, when combined with differentially private mechanisms, discarding the magnitude is expected to improve the privacy guarantees. Specifically, we consider the noisy sign-based compressor combined with Gaussian noise, denoted by G-NoisySign, given in Algorithm 1. The G-NoisySign com-

---

**Algorithm 1** G-NoisySign Compressor (Chen et al., 2020b)

**Input**: $c > 0$, $x \in [-c, c]$, Gaussian noise $n$ with zero mean and variance $\sigma_{DP}^2$.
Privatization: $Z \triangleq sign(x + n)$.

---

pressor first perturbs the input $x$ with a Gaussian noise $n$ with zero mean and a variance of $\sigma_{DP}^2$ and then returns the sign of the noisy result. According to (Dong et al., 2021), it is known that directly releasing $x + n$ gives $\mu$-GDP with $\mu = \frac{2c}{\sigma_{DP}}$, which further suggests $(\epsilon, \delta)$-DP with $\delta(\epsilon) = \Phi(-\frac{\epsilon}{\mu} + \frac{\mu}{2}) - e^\epsilon\Phi(-\frac{\epsilon}{\mu} - \frac{\mu}{2})$ for any $\epsilon \geq 0$. If we measure the $\mu$-GDP and $(\epsilon, \delta)$-DP in terms of tradeoff functions, we have $f_\mu^{Gaussian}(\alpha) = \Phi(\Phi^{-1}(1 - \alpha) - \mu)$ and $f_{\epsilon,\delta}^{Gaussian}(\alpha) = \max\{0, 1 - \delta - e^\epsilon\alpha, e^{-\epsilon}(1 - \delta - \alpha)\}$, respectively. In the following, we prove that releasing $sign(x + n)$ instead of $x + n$ adds another level of privacy protection. Utilizing Theorem 1, we readily obtain the $f$-DP guarantee of G-NoisySign as follows.

$$
f^{GNS}(\alpha)
$$
$$
= \begin{cases} 1 - \frac{\Phi(\frac{c}{\sigma_{DP}})}{\Phi(-\frac{c}{\sigma_{DP}})}\alpha, & \text{for } \alpha \in [0, \Phi(-\frac{c}{\sigma_{DP}})], \\ \frac{\Phi(-\frac{c}{\sigma_{DP}})}{\Phi(\frac{c}{\sigma_{DP}})} - \frac{\Phi(-\frac{c}{\sigma_{DP}})}{\Phi(\frac{c}{\sigma_{DP}})}\alpha, & \text{for } \alpha \in [\Phi(-\frac{c}{\sigma_{DP}}), 1]. \end{cases}
$$
(12)

Figure 1 compares the privacy guarantees of the G-NoisySign mechanism with the Gaussian mechanism (i.e., directly releasing $x+n$) given $c = 1$ and $\sigma_{DP} = 2$. It can be observed that $f^{GNS}(\alpha) \geq f_\mu^{Gaussian}(\alpha) \geq f_{\epsilon,\delta}^{Gaussian}(\alpha)$, i.e., given the same type I error rate $\alpha$, G-NoisySign attains a larger type II error rate than the vanilla Gaussian mechanism, which suggests that it is more difficult for the adversary to distinguish the inputs, or equivalently, better privacy.

**Logistic Mechanism:** Note that the generic sign-based compressor in Definition 4 captures the logistic mechanism in (Jang et al., 2024) as a special case. More specifically, the logistic mechanism replaces the Gaussian noise $n$ in

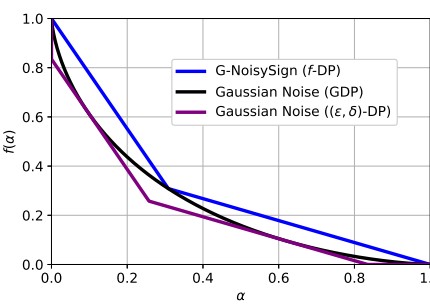

*Figure 1.* Privacy Amplification of $sign(\cdot)$. For "G-NoisySign", the tradeoff function is given by (12) with $c = 1$ and $\sigma_{DP} = 2$; for "Gaussian Noise (GDP)", the tradeoff function is given by $f_\mu^{Gaussian}(\alpha)$ with $\mu = 1$; for "Gaussian Noise (($\epsilon, \delta$)-DP)", the tradeoff function is given by $f_{\epsilon,\delta}^{Gaussian}(\alpha)$ with $\epsilon = \ln(\frac{\Phi(\frac{c}{\sigma_{DP}})}{\Phi(-\frac{c}{\sigma_{DP}})})$ and $\delta = \Phi(-\frac{\epsilon}{\mu} + \frac{\mu}{2}) - e^\epsilon \Phi(-\frac{\epsilon}{\mu} - \frac{\mu}{2})$, in which $c = \mu = 1$ and $\sigma_{DP} = 2$.

Algorithm 1 with a logistic noise $n \sim Logistic(0, s)$ for some hyperparameter $s$, which leads to $p_{+1} = \frac{e^{\frac{x}{2s}}}{e^{\frac{x}{2s}} + e^{-\frac{x}{2s}}}$ and $p_{-1} = \frac{e^{-\frac{x}{2s}}}{e^{\frac{x}{2s}} + e^{-\frac{x}{2s}}}$, respectively. Utilizing Theorem 1 yields the following $f$-DP guarantee for L-NoisySign.

$$f^{LNS}(\alpha) = \begin{cases} 1 - e^{\frac{c}{s}}\alpha, & \text{for } \alpha \in [0, \frac{e^{-\frac{c}{2s}}}{e^{\frac{c}{2s}} + e^{-\frac{c}{2s}}}], \\ e^{-\frac{c}{s}} - e^{-\frac{c}{s}}\alpha, & \text{for } \alpha \in [\frac{e^{-\frac{c}{2s}}}{e^{\frac{c}{2s}} + e^{-\frac{c}{2s}}}, 1]. \end{cases}$$

(13)

**Comparison between G-NoisySign and L-NoisySign**: It can be readily shown that L-NoisySign attains the same privacy guarantees as G-NoisySign when $e^{\frac{c}{s}} = \frac{\Phi(\frac{c}{\sigma_{DP}})}{\Phi(-\frac{c}{\sigma_{DP}})}$, which means $s = c / \ln(\Phi(\frac{c}{\sigma_{DP}})/\Phi(-\frac{c}{\sigma_{DP}}))$. While (Jang et al., 2024) suggests that the logistic mechanism outperforms the Gaussian mechanism when the $sign(\cdot)$ compressor is applied, we find that this is because the privacy amplification effect of discarding the magnitude is ignored. More specifically, in the following, we show that G-NoisySign may be better than L-NoisySign in terms of the estimation error in distributed mean estimation. Particularly, utilizing Taylor expansion, we have

$$\mathbb{E}[sign(x + \mathcal{N}(0, \sigma_{DP}))] = \Phi\left(\frac{x}{\sigma_{DP}}\right) - \Phi\left(-\frac{x}{\sigma_{DP}}\right)$$

$$= \frac{1}{\sqrt{2\pi}} \int_{-\frac{x}{\sigma_{DP}}}^{\frac{x}{\sigma_{DP}}} e^{-\frac{t^2}{2}} dt = \frac{1}{\sqrt{2\pi}} \int_{-\frac{x}{\sigma_{DP}}}^{\frac{x}{\sigma_{DP}}} \sum_{k=0}^\infty (-1)^k \frac{x^{2k}}{2^k k!} dt$$

$$= \frac{2}{\sqrt{2\pi}\sigma_{DP}} x - \frac{1}{3\sqrt{2\pi}\sigma_{DP}^3} x^3 + \mathcal{O}\left(\frac{x^5}{\sigma_{DP}^5}\right),$$

(14)

and

$$\mathbb{E}[sign(x + Logistic(0, s))] = \frac{1}{2}(\tanh(\frac{x}{2s}) - \tanh(-\frac{x}{2s}))$$

$$= \frac{x}{2s} - \frac{x^3}{24s^3} + \mathcal{O}\left(\frac{x^5}{s^5}\right).$$

(15)

Therefore, we use $\frac{\sqrt{2\pi}\sigma_{DP}}{2} \times sign(x + \mathcal{N}(0, \sigma_{DP}))$ and $2s \times sign(x + Logistic(0, s))$ as the estimate of $x$ for G-NoisySign and L-NoisySign, respectively. In the high-privacy regime (i.e., large $\sigma_{DP}$ and $s$), the estimation errors $|\Delta_G|$ and $|\Delta_L|$ are dominated by the second terms in (14) and (15), respectively, which are given by

$$|\Delta_G| = \frac{|x|^3}{6\sigma_{DP}^2}, \quad |\Delta_L| = \frac{|x|^3}{12s^2}.$$

(16)

When $s = c / \ln(\Phi(\frac{c}{\sigma_{DP}})/\Phi(-\frac{c}{\sigma_{DP}}))$, the sufficient and necessary condition for $|\Delta_G| < |\Delta_L|$ is given by

$$\ln\left(\frac{\Phi(\frac{c}{\sigma_{DP}})}{\Phi(-\frac{c}{\sigma_{DP}})}\right) > \frac{\sqrt{2}c}{\sigma_{DP}}.$$

(17)

Let $h(x) = \ln(\frac{\Phi(x)}{\Phi(-x)}) - \sqrt{2}x$, then $h'(x) = \frac{\phi(x)}{\Phi(x)(1-\Phi(x))} - \sqrt{2} = \frac{\phi(x)}{\Phi(x)} + \frac{\phi(-x)}{\Phi(-x)} - \sqrt{2}$, in which $\phi(\cdot)$ is the probability density function of the standard normal distribution. $\frac{\phi(x)}{\Phi(x)}$ is the famous inverse mills ratio for the standard normal distribution, which is known to be strictly convex (Gasull & Utzet, 2014), which suggests $\frac{\phi(x)}{\Phi(x)} + \frac{\phi(-x)}{\Phi(-x)} > 2\frac{\phi(0)}{\Phi(0)} = \frac{4}{\sqrt{2\pi}} > \sqrt{2}$. Therefore, $h(x) > h(0) = 0$ for $x > 0$, i.e., the inequality (17) holds for any $c, \sigma_{DP} > 0$. As a result, we conclude that, in the high-privacy regime, G-NoisySign is more suitable if we are more concerned with the estimation error. This corresponds to the distributed mean estimation scenario, which is the building block of federated learning.

Figure 2 presents numerical results that compare G-NoisySign and L-NoisySign in terms of the ratios between the estimation error (i.e., $|\Delta_G|$ and $|\Delta_L|$) and the input magnitude $|x|$ for different $\sigma_{DP}$ and $x$, which validate the analytical results above.

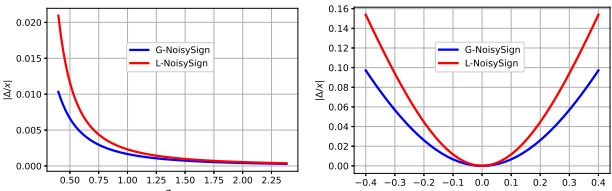

*Figure 2.* The comparison of the ratios between the estimation error and the input magnitude for G-NoisySign and L-NoisySign. For the left figure, we fix $c = 1$ and $x = 0.1$ with varying $\sigma_{DP}$, while for the right figure, we fix $c = 1$ and $\sigma_{DP} = 0.5$ with varying $x$. For L-NoisySign, we set $s = c / \ln(\Phi(\frac{c}{\sigma_{DP}})/\Phi(-\frac{c}{\sigma_{DP}}))$ to ensure the same privacy guarantee.

**Remark 1.** *(Jang et al., 2024) considers the error rate of sign sampling as the performance metric. We note that this may not be a good performance indicator for federated learning with heterogeneous data across workers. In particular, it has been observed that the vanilla SIGNSGD, which corresponds to the case that the error rate of sign sampling is 0, fails to converge in the presence of data heterogeneity (Karimireddy et al., 2019; Chen et al., 2020b; Jin et al.,*

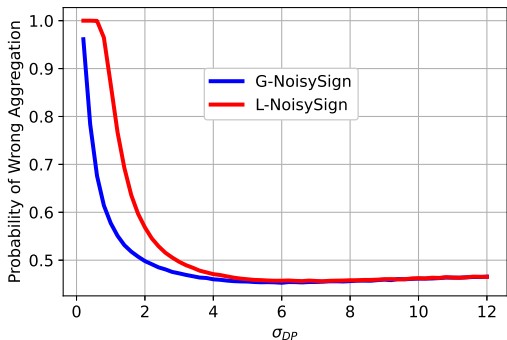

*Figure 3.* The comparison of the probability of wrong aggregation between G-NoisySign and L-NoisySign. A set of 100 workers are considered, with $x_m = -0.05$ for $1 \leq m \leq 98$ and $x_m = 10$ for $99 \leq m \leq 100$. We set $c = 10$.

*2024). With such consideration, we present additional numerical results that evaluate the probability of wrong aggregation of G-NoisySign and L-NoisySign in Figure 3. More specifically, we consider a set of workers $m \in \mathcal{H}$, each with some data $x_m$, and we are interested in the probability of wrong aggregation, i.e., $P(sign(\frac{1}{M} \sum_{m \in \mathcal{H}} sign(x_m + n)) \neq sign(\frac{1}{M} \sum_{m \in \mathcal{H}} x_m))$. It can be observed in Figure 3 that, in the presence of data heterogeneity, G-NoisySign may outperform L-NoisySign and attain a lower probability of wrong aggregation. Note that when $\sigma_{DP}$ is small, the probability of wrong aggregation is larger than 0.5, which confirms the results that SIGNSGD may fail with heterogeneous data. As $\sigma_{DP}$ increases, the probability of wrong aggregation first decreases and then increases. Considering that SIGNSGD converges to a stationary point when the probability of wrong aggregation is less than 0.5 (Safaryan & Richtárik, 2021), this implies that adding appropriate noise may benefit both privacy and aggregation accuracy. However, it does not mean that G-NoisySign always outperforms L-NoisySign. More numerical results concerning the impact of data heterogeneity can be found in Appendix C.2.*

### 4.3. Extending to the Vector Case

The privacy analyses in Section 4.1 are concerned with the scalar case, which is extended to the vector case in this section. More specifically, we consider a higher dimensional hypothesis testing problem as in (Zheng et al., 2020) and (Dong et al., 2021), and the most powerful test is also threshold-based according to the Neyman-Pearson lemma.

**Theorem 2.** *For any vectors $\boldsymbol{x}$ and $\boldsymbol{x}'$ with $\|\boldsymbol{x}\|_2, \|\boldsymbol{x}'\|_2 \leq C$, the tradeoff function $f^{GNS}(\alpha)$ for the corresponding hypothesis testing problem satisfies*

$$f^{GNS}(\alpha) \xrightarrow{d \to \infty} G_\mu(\alpha) = \Phi(\Phi^{-1}(1-\alpha) - \mu), \quad (18)$$

*in which $\mu$ is the limit of $\mu_d$ with*

$$\mu_d \leq \frac{2C}{\sqrt{2\pi \Phi\left(\frac{C}{\sqrt{d}\sigma}\right) \Phi\left(-\frac{C}{\sqrt{d}\sigma}\right)}\sigma}. \quad (19)$$

*Sketch of Proof.* In the vector case, we consider a higher dimensional hypothesis testing problem given by

$$H_0 : \boldsymbol{z} \sim P = P_1 \times P_2 \times \cdots \times P_d,$$
$$H_1 : \boldsymbol{z} \sim Q = Q_1 \times Q_2 \times \cdots \times Q_d. \quad (20)$$

Let $T_d(\boldsymbol{z}) = \log\left(\frac{\prod_{i=1}^d q_i(\boldsymbol{z}_i)}{\prod_{i=1}^d p_i(\boldsymbol{z}_i)}\right)$ denote the privacy loss, in which $\boldsymbol{z} = [\boldsymbol{z}_1, \boldsymbol{z}_2, ..., \boldsymbol{z}_d]$ is the output of the mechanism, $p_i(\cdot)$ and $q_i(\cdot)$ denote the probability density functions of $P_i$ and $Q_i$ given the input vectors $\boldsymbol{x}$ and $\boldsymbol{x}'$, respectively. Similar to (Jang et al., 2024), the maximizer of the privacy loss is given by $\boldsymbol{x}' = -\boldsymbol{x} = \frac{C}{\sqrt{d}} \cdot \mathbf{1}$.

According to the Neyman-Pearson lemma (Lehmann et al., 2005), the most powerful test at level $\alpha$ is a thresholding function of $T_d(\boldsymbol{z})$, and we have

$$f(\alpha) \to G_\mu(\alpha) = \Phi(\Phi^{-1}(1-\alpha) - \mu), \quad (21)$$

in which $\mu$ is the limit of $\mu_d$ with

$$\mu_d = \frac{\mathbb{E}_Q[T_d(\boldsymbol{z})] - \mathbb{E}_P[T_d(\boldsymbol{z})]}{\sqrt{\mathbb{V}\text{ar}_Q[T_d(\boldsymbol{z})]}}. \quad (22)$$

For G-NoisySign, we have
$\mathbb{E}_Q[T_d(\boldsymbol{z})] - \mathbb{E}_P[T_d(\boldsymbol{z})]$

$$= \sum_i^d \left[\Phi\left(\frac{\boldsymbol{x}_i'}{\sigma}\right) - \Phi\left(\frac{\boldsymbol{x}_i}{\sigma}\right)\right] \log\left(\frac{\Phi\left(\frac{\boldsymbol{x}_i'}{\sigma}\right)\Phi\left(-\frac{\boldsymbol{x}_i}{\sigma}\right)}{\Phi\left(-\frac{\boldsymbol{x}_i'}{\sigma}\right)\Phi\left(\frac{\boldsymbol{x}_i}{\sigma}\right)}\right)$$

$$= 2d\left[\Phi\left(\frac{C}{\sqrt{d}\sigma}\right) - \Phi\left(-\frac{C}{\sqrt{d}\sigma}\right)\right] \log\left(\frac{\Phi\left(\frac{C}{\sqrt{d}\sigma}\right)}{\Phi\left(-\frac{C}{\sqrt{d}\sigma}\right)}\right), \quad (23)$$

and
$\mathbb{V}\text{ar}_Q[T_d(\boldsymbol{z})]$

$$= \sum_i^d \left[\Phi\left(\frac{\boldsymbol{x}_i'}{\sigma}\right)\Phi\left(-\frac{\boldsymbol{x}_i'}{\sigma}\right)\right] \log^2\left(\frac{\Phi\left(\frac{\boldsymbol{x}_i'}{\sigma}\right)\Phi\left(-\frac{\boldsymbol{x}_i}{\sigma}\right)}{\Phi\left(-\frac{\boldsymbol{x}_i'}{\sigma}\right)\Phi\left(\frac{\boldsymbol{x}_i}{\sigma}\right)}\right)$$

$$= 4d\left[\Phi\left(\frac{C}{\sqrt{d}\sigma}\right)\Phi\left(-\frac{C}{\sqrt{d}\sigma}\right)\right] \log^2\left(\frac{\Phi\left(\frac{C}{\sqrt{d}\sigma}\right)}{\Phi\left(-\frac{C}{\sqrt{d}\sigma}\right)}\right). \quad (24)$$

As a result,

$$\mu_d = \frac{\mathbb{E}_Q[T_d(\boldsymbol{z})] - \mathbb{E}_P[T_d(\boldsymbol{z})]}{\sqrt{\mathbb{V}\text{ar}_Q[T_d(\boldsymbol{z})]}}$$

$$= \frac{\left[\Phi\left(\frac{C}{\sqrt{d}\sigma}\right) - \Phi\left(-\frac{C}{\sqrt{d}\sigma}\right)\right]\sqrt{d}}{\sqrt{\Phi\left(\frac{C}{\sqrt{d}\sigma}\right)\Phi\left(-\frac{C}{\sqrt{d}\sigma}\right)}}. \quad (25)$$

The mean value theorem (Rudin et al., 1964) implies that

$$\left|\Phi\left(\frac{\boldsymbol{x}_i'}{\sigma}\right) - \Phi\left(\frac{\boldsymbol{x}_i}{\sigma}\right)\right| \leq \frac{1}{\sigma\sqrt{2\pi}}|\boldsymbol{x}_i' - \boldsymbol{x}_i|. \quad (26)$$

**Algorithm 2** Differentially Private Noisy sɪɢɴSGD

---

**Initialization**: The initial model weights $\boldsymbol{w}_0$, the batch size $b$, the clipping function **Clip**$(\cdot)$, the learning rate $\eta$, and the total number of communication rounds $T$.

**for** communication round $t = 0, 1, 2, \cdots, T$ **do**

The central server sends the model weights $\boldsymbol{w}^{(t)}$ to the workers.

**for** each worker $i \in \mathcal{H}$ **do**

Sample a mini-batch $\mathcal{B}_i^{(t)}$ training examples of size $b$ at random from $\mathcal{D}_i$. Then compute and clip the per-example gradients and average the mini-batch gradients:

$$\boldsymbol{g}_i^{(t)} = \frac{1}{b} \sum_{s \in \mathcal{B}_i^{(t)}} \textbf{Clip}(\nabla F_i(\boldsymbol{w}^{(t)}; s)) \qquad (28)$$

Compress and send $\boldsymbol{Z}_i^{(t)} = \mathcal{C}(\boldsymbol{g}_i^{(t)}) = sign(\boldsymbol{g}_i^{(t)} + \mathcal{N}(0, \sigma_{DP}^2 \cdot \boldsymbol{I}))$ back to the central server.

**end for**

The central server aggregates the received gradients with two possible schemes

$$\hat{\boldsymbol{g}}_A^{(t)} = Agg(\{\boldsymbol{Z}_i^{(t)}\}_{i \in \mathcal{H}})$$

$$= \begin{cases} \frac{1}{|\mathcal{H}|} \sum_{i \in \mathcal{H}} \boldsymbol{Z}_i^{(t)}, & \text{Scheme I,} \\ sign\left(\frac{1}{|\mathcal{H}|} \sum_{i \in \mathcal{H}} \boldsymbol{Z}_i^{(t)}\right), & \text{Scheme II.} \end{cases} \quad (29)$$

The central server updates the model by

$$\boldsymbol{w}^{(t+1)} = \boldsymbol{w}^{(t)} - \eta \hat{\boldsymbol{g}}_A^{(t)}. \qquad (30)$$

**end for**

---

Therefore,

$$\mu_d \leq \frac{2C}{\sqrt{2\pi \Phi\left(\frac{C}{\sqrt{d}\sigma}\right) \Phi\left(-\frac{C}{\sqrt{d}\sigma}\right)}\sigma}, \qquad (27)$$

which completes the proof. □

**Remark 2.** *We note that the Gaussian mechanism with variance $\sigma$ has a $\mu_G$-GDP guarantee with $\mu_G = 2C/\sigma$ (Dong et al., 2019). In Theorem 2, when $d$ and $\sigma$ increase (which corresponds to a larger neural network and more stringent privacy requirement), the privacy guarantee of G-NoisySign converges to $\mu_{GNS} = 2C/\sigma\sqrt{\pi/2}$. Therefore, there is an improvement by a factor of $\sqrt{\pi/2}$. Utilizing the composition property in Lemma 2, for the same overall privacy guarantee, G-NoisySign allows for $\pi/2$ more training steps than the classic Gaussian mechanism. Therefore, while (Jang et al., 2024) claims a similar improvement (i.e., $1.5\times$ more training steps) when utilizing the L-NoisySign mechanism, our result suggests that this is mainly due to the fact that the privacy amplification effect of the $sign(\cdot)$ compressor over the Gaussian mechanism is ignored.*

## 5. Convergence of Noisy sɪɢɴSGD

In this section, we show the convergence of Noisy sɪɢɴSGD

with the G-NoisySign compressor, i.e., Algorithm 2. Different from (Jang et al., 2024) which mainly considers centralized learning, we focus on distributed learning with data heterogeneity. To facilitate the convergence analysis, we make the following commonly adopted assumptions.

**Assumption 1.** *(Lower bound). For all $\boldsymbol{x}$ and some constant $F^*$, we have objective value $F(\boldsymbol{x}) \geq F^*$.*

**Assumption 2.** *(Smoothness). $\forall \boldsymbol{y}, \boldsymbol{x}$, we require for some non-negative constant $L$,*

$$F(\boldsymbol{y}) \leq F(\boldsymbol{x}) + \langle \nabla F(\boldsymbol{x}), \boldsymbol{y} - \boldsymbol{x} \rangle + \frac{L}{2}\|\boldsymbol{y} - \boldsymbol{x}\|_2^2, \quad (31)$$

*where $\langle \cdot, \cdot \rangle$ is the standard inner product.*

**Assumption 3.** *(Variance bound). For any worker $m$, the stochastic gradient oracle gives an independent unbiased estimate $\boldsymbol{g}_m$ that has coordinate-wise bounded variance:*

$$\mathbb{E}[\boldsymbol{g}_m] = \nabla F_m(\boldsymbol{w}), \mathbb{E}[(\boldsymbol{g}_{m,i} - \nabla F_m(\boldsymbol{w})_i)^2] \leq \sigma_{L,i}, \quad (32)$$

*for a vector of non-negative constants $\bar{\boldsymbol{\sigma}}_L = [\sigma_{L,1}, \cdots, \sigma_{L,d}]$.*

**Assumption 4.** *(Gradient bound). For any worker $m$, the stochastic gradient satisfies $|\boldsymbol{g}_{m,i}| \leq c, \forall 1 \leq i \leq d$.*

Note that clipping is usually applied to ensure bounded gradients in DP-SGD (Abadi et al., 2016). In this work, we follow the literature (e.g., (Chen et al., 2020b; Xiang & Su, 2023)) and adopt the bounded gradient assumption, i.e., Assumption 4, for convergence analysis. The impact of gradient clipping is left for future work.

**Theorem 3** (**Convergence of Noisy sɪɢɴSGD with Scheme I**). *Suppose Assumptions 1-4 are satisfied, and the learning rate is set as $\eta = \frac{1}{\sqrt{TLd}}$. Then by running Algorithm 2 for $T$ iterations, we have*

$$\frac{1}{T}\sum_{t=1}^{T}\left\|\nabla F(\boldsymbol{w}^{(t)})\right\|_2^2 \leq \frac{(F(\boldsymbol{w}^{(0)}) - F^*)\sqrt{Ld}\sqrt{2\pi}\sigma_{DP}}{\sqrt{T}}$$
$$+ \frac{\sqrt{Ld}\sqrt{2\pi}\sigma_{DP}}{2\sqrt{T}} + \mathcal{O}\left(\frac{c^6}{\sigma_{DP}^4}\right). \tag{33}$$

**Remark 3.** *Adding a Gaussian noise with variance $\sigma_{DP}$ yields $\mu$-GDP with $\mu = \mathcal{O}(1/\sigma_{DP})$, and the composition of $\mu$-GDP mechanisms in Lemma 2 suggests an overall privacy guarantee $\mu_T = \mathcal{O}\left(\sqrt{T}/\sigma_{DP}\right)$ over $T$ communication rounds. If we set $\sigma_{DP} = \mathcal{O}\left(\sqrt{T}/\mu_T\right)$, Theorem 3 implies $\frac{1}{T}\sum_{t=1}^{T}\left\|\nabla F(\boldsymbol{w}^{(t)})\right\|_2^2 \leq \mathcal{O}(1/\mu_T) + \mathcal{O}\left(c^6\mu_T^4/T^2\right)$, i.e., it converges to the neighborhood of the local optimum with a gap $\mathcal{O}(1/\mu_T)$, which matches the state-of-the-art results (Fang et al., 2022; Koloskova et al., 2023).*

**Theorem 4** (**Convergence of Noisy sɪɢɴSGD with Scheme II**). *Suppose Assumptions 1-4 are satisfied, and the learning rate is set as $\eta = \frac{1}{\sqrt{TLd}}$. Then by running Algorithm 2 for*

*Table 1.* Fashion-MNIST Test Accuracy for Varying Privacy Requirements Per Communication Round ($\alpha = 0.1$)

| $\mu$ | 0.04 | 0.08 | 0.4 | 0.8 | 1.6 |
|---|---|---|---|---|---|
| GAUSSIAN NOISE | $43.78 \pm 1.98\%$ | $58.69 \pm 1.48\%$ | $73.48 \pm 0.49\%$ | $77.17 \pm 0.51\%$ | $79.90 \pm 0.25\%$ |
| G-NOISYSIGN | $45.24 \pm 2.51\%$ | $58.46 \pm 2.48\%$ | $73.78 \pm 0.57\%$ | $77.20 \pm 0.29\%$ | $79.57 \pm 0.57\%$ |
| G-NOISYSIGN-VOTE | $42.18 \pm 4.03\%$ | $56.59 \pm 1.91\%$ | $73.23 \pm 0.66\%$ | $76.47 \pm 0.61\%$ | $79.27 \pm 0.37\%$ |
| L-NOISYSIGN | $42.52 \pm 3.11\%$ | $58.39 \pm 1.53\%$ | $73.89 \pm 0.78\%$ | $77.18 \pm 0.31\%$ | $79.66 \pm 0.45\%$ |
| L-NOISYSIGN-VOTE | $40.46 \pm 3.82\%$ | $53.89 \pm 1.82\%$ | $73.23 \pm 0.47\%$ | $76.49 \pm 0.31\%$ | $79.24 \pm 0.31\%$ |

*Table 2.* CIFAR-10 Test Accuracy for Varying Privacy Requirements Per Communication Round ($\alpha = 1$)

| $\mu$ | 0.8 | 1.6 | 4 | 8 | 16 |
|---|---|---|---|---|---|
| GAUSSIAN NOISE | $40.77 \pm 1.22\%$ | $48.02 \pm 0.69\%$ | $56.73 \pm 0.60\%$ | $63.13 \pm 1.15\%$ | $67.53 \pm 0.58\%$ |
| G-NOISYSIGN | $40.28 \pm 0.66\%$ | $47.80 \pm 0.97\%$ | $56.56 \pm 1.10\%$ | $62.70 \pm 1.57\%$ | $66.82 \pm 1.22\%$ |
| G-NOISYSIGN-VOTE | $38.10 \pm 1.24\%$ | $46.87 \pm 0.84\%$ | $55.39 \pm 1.03\%$ | $61.02 \pm 0.96\%$ | $65.70 \pm 1.14\%$ |
| L-NOISYSIGN | $40.06 \pm 0.81\%$ | $47.77 \pm 0.62\%$ | $56.72 \pm 1.29\%$ | $62.46 \pm 1.38\%$ | $66.69 \pm 1.07\%$ |
| L-NOISYSIGN-VOTE | $37.46 \pm 0.84\%$ | $46.33 \pm 0.70\%$ | $55.53 \pm 0.90\%$ | $61.27 \pm 1.21\%$ | $65.25 \pm 0.71\%$ |

*T iterations, we have*

$$\frac{1}{T} \sum_{t=1}^{T} ||\nabla F(\boldsymbol{w}^{(t)})||_1 \leq \frac{(F(\boldsymbol{w}^{(0)}) - F^*)\sqrt{Ld}}{\sqrt{T}} + \frac{\sqrt{Ld}}{2\sqrt{T}}$$

$$+ \left[ \frac{\sigma_{DP} d\sqrt{2\pi}}{\sqrt{M}} + \frac{2||\bar{\boldsymbol{\sigma}}_L||_1}{\sqrt{M}} + \mathcal{O}\left(\frac{c^3}{\sigma_{DP}^2}\right) \right]$$

$$\leq \mathcal{O}\left(\frac{1}{\sqrt{T}}\right) + \mathcal{O}\left(\frac{\sigma_{DP}}{\sqrt{M}} + \frac{||\bar{\boldsymbol{\sigma}}_L||_1}{\sqrt{M}}\right) + \mathcal{O}\left(\frac{c^3}{\sigma_{DP}^2}\right).$$

$$(34)$$

**Remark 4.** *The bound in Theorem 4 depends on $M$. When $\sigma_{DP} = \mathcal{O}(\sqrt{T}/\mu_T)$ and $M = \mathcal{O}(T)$ (note that $M$ is usually large in federated learning), we obtain a convergence rate of $\frac{1}{T} \sum_{t=1}^{T} ||\nabla F(\boldsymbol{w}^{(t)})||_1 \leq \mathcal{O}(1/\mu_T) + \mathcal{O}\left(1/\sqrt{T}\right)$.*

## 6. Experimental Results

In this section, we present experimental results to validate our theoretical analyses in the previous sections.

**Datasets and Models**: We evaluate the performance of the algorithms on two commonly used benchmarks for differentially private distributed learning: Fashion-MNIST (Xiao et al., 2017) and CIFAR-10 (Krizhevsky et al., 2009). We adopt a three-layer fully connected neural network for Fashion-MNIST and the DPNASNet-CIFAR (Cheng et al., 2022) for CIFAR-10. For Fashion-MNIST, we consider a scenario of $M = 100$ workers with the training data on each worker drawn independently with class labels following a Dirichlet distribution $Dir(\alpha)$ with $\alpha = 0.1$, and 50 workers are sampled uniformly at random for training during each communication round. For CIFAR-10, we consider a scenario of $M = 30$ workers with $\alpha = 1$, and all workers participate in training during each communication round. We train both neural networks from scratch with a batch

size of 32 in our experiments.

**Hyperparameters**: The per-example gradient clipping thresholds are set to $C = 1$ and $C = 2$ for Fashion-MNIST and CIFAR-10, respectively. We tune the learning rate from the set $\{0.001, 0.002, 0.003, 0.005, 0.01, 0.02, 0.03, 0.05, 0.1, 0.2, 0.3, 0.5, 1, 2, 3, 5, 10\}$ and run the algorithms for 500 communication rounds. We run all the algorithms for 10 repeats and present the mean test accuracy.

**Results**: We compare the Noisy SIGNSGD algorithms with the G-NoisySign compressor and the L-NoisySign compressor with the classic Gaussian mechanism (i.e., DP-SGD (Abadi et al., 2016)). Note that the Gaussian mechanism has been shown to be order-optimal (without considering the communication efficiency) theoretically (Cai et al., 2021) and achieve the state-of-the-art privacy-utility tradeoff experimentally (Guo et al., 2023). Table 1 and Table 2 show that Noisy SIGNSGD with G-NoisySign and L-NoisySign attain comparable performance to the classic Gaussian mechanism. In this sense, despite that discarding the magnitude information by adopting the $sign(\cdot)$ compressor may result in performance degradation, it leads to enhanced privacy and achieves a comparable privacy-utility tradeoff. Note that Noisy SIGNSGD requires only 1 bit for each coordinate of the gradients and, therefore, provides an improvement of $32\times$ in communication efficiency. The results for the convergence over communication rounds are presented in Appendix C.2.2.

It is also worth mentioning that employing majority vote on the server side leads to performance degradation (less than 2% in most of the examined scenarios). While the local privacy guarantees remain the same since it is independent of the aggregation strategy on the server side, there is supposed to be an enhancement in central differential privacy guaran-

tees since the server releases less information by adopting another $sign(\cdot)$ compressor (i.e., scheme II in (29)). Investigating the privacy amplification effect of the majority vote mechanism remains an interesting future direction.

## 7. Conclusion

In this paper, we investigate the privacy amplification effect of the $sign(\cdot)$ compressor, based on which we further show that the Noisy SIGN SGD algorithm achieves a convergence rate and privacy-utility trade-off comparable to the classic DP-SGD both theoretically and experimentally. The analyses and results are expected to shed light on the development of privacy-preserving and communication-efficient distributed learning algorithms.

## Acknowledgements

Richeng Jin is supported in part by the National Key R&D Program of China under Grant No. 2024YFE0200804, in part by the National Natural Science Foundation of China under Grant No. 62301487, in part by the Zhejiang Provincial Natural Science Foundation of China under Grant No. LQ23F010021, and in part by Information Technology Center and State Key Lab of CAD & CG, Zhejiang University. Huaiyu Dai is supported by the US National Science Foundation under Grant ECCS-2203214. The views expressed in this publication are those of the authors and do not necessarily reflect the views of the National Science Foundation. The authors also appreciate the constructive comments provided by the anonymous reviewers.

## Impact Statement

This paper presents work whose goal is to advance the field of privacy-preserving machine learning. We believe that the proposed tool and method will benefit the design of communication-efficient and differentially private distributed learning algorithms. We do not find any potential negative societal consequences of this work that must be specifically highlighted here.

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

# A. Proofs

## A.1. Proof of Theorem 1

**Lemma 3.** *(Neyman-Pearson Lemma (Lehmann et al., 2005)) Let $P$ and $Q$ be probability distributions on $\Omega$ with densities $p$ and $q$, respectively. For the hypothesis testing problem $H_0 : P$ vs $H_1 : Q$, a test $\phi : \Omega \to [0,1]$ is the most powerful test at level $\alpha$ if and only if there are two constants $h \in [0, +\infty]$ and $\gamma \in [0,1]$ such that $\phi$ has the form*

$$\phi(x) = \begin{cases} 1, \text{ if } \frac{q(x)}{p(x)} > h, \\ \gamma, \text{ if } \frac{q(x)}{p(x)} = h, \\ 0, \text{ if } \frac{q(x)}{p(x)} < h, \end{cases} \tag{35}$$

*and $\mathbb{E}_P[\phi] = \alpha$. The rejection rule suggests that $H_0$ is rejected with a probability of $\phi(x)$ given the observation $x$.*

In the following, we show the $f$-DP of the sign-based compressor.

**Theorem 1.** *Suppose that $p_{max} + p_{min} = 1$, and $p_{+1}(x) > p_{+1}(y), \forall x > y$. The sign-based compressor is $f^{NoisySign}(\alpha)$-differentially private with*

$$f^{NoisySign}(\alpha) = \begin{cases} 1 - \frac{p_{max}}{p_{min}}\alpha, & \text{for } \alpha \in [0, p_{min}], \\ \frac{p_{min}}{p_{max}} - \frac{p_{min}}{p_{max}}\alpha, & \text{for } \alpha \in [p_{min}, 1]. \end{cases} \tag{36}$$

*Proof.* Let $Y = NoisySign(x', p_{+1}, p_{-1})$ and $X = NoisySign(x, p_{+1}, p_{-1})$, we have

$$\frac{P(Y=-1)}{P(X=-1)} = \frac{p_{-1}(x')}{p_{-1}(x)},$$

$$\frac{P(Y=1)}{P(X=1)} = \frac{p_{+1}(x')}{p_{+1}(x)}. \tag{37}$$

When $x > x'$, it can be observed that $\frac{P(Y=-1)}{P(X=-1)} > 1 > \frac{P(Y=1)}{P(X=1)}$. In this case, when $\alpha \in [0, p_{-1}(x)]$, we set $h = \frac{P(Y=-1)}{P(X=-1)}$ in Lemma 3, and therefore

$$\mathbb{E}_P[\phi] = \gamma P(X=-1) = \gamma p_{-1}(x) = \alpha, \tag{38}$$

and

$$\beta_\phi^{x>x'}(\alpha) = 1 - \mathbb{E}_Q[\phi] = 1 - \gamma P(Y=-1) = 1 - \gamma p_{-1}(x') = 1 - \frac{p_{-1}(x')}{p_{-1}(x)}\alpha. \tag{39}$$

When $\alpha \in [p_{-1}(x), 1]$, we set $h = \frac{P(Y=1)}{P(X=1)}$ in Lemma 3, and therefore

$$\mathbb{E}_P[\phi] = P(X=-1) + \gamma P(X=1) = p_{-1}(x) + \gamma p_{+1}(x) = \alpha, \tag{40}$$

and

$$\beta_\phi^{x>x'}(\alpha) = 1 - \mathbb{E}_Q[\phi] = 1 - P(Y=-1) - \gamma P(Y=1) = p_{+1}(x') - \gamma p_{+1}(x') = p_{+1}(x') - \frac{\alpha - p_{-1}(x)}{p_{+1}(x)}p_{+1}(x'). \tag{41}$$

In summary, we have

$$\beta_\phi^{x>x'}(\alpha) = \begin{cases} 1 - \frac{p_{-1}(x')}{p_{-1}(x)}\alpha, & \text{for } \alpha \in [0, p_{-1}(x)], \\ \frac{p_{+1}(x')}{p_{+1}(x)} - \frac{p_{+1}(x')}{p_{+1}(x)}\alpha, & \text{for } \alpha \in [p_{-1}(x), 1]. \end{cases} \tag{42}$$

When $x < x'$, it can be observed that $\frac{P(Y=-1)}{P(X=-1)} < 1 < \frac{P(Y=1)}{P(X=1)}$. In this case, when $\alpha \in [0, p_{+1}(x)]$, we set $h = \frac{P(Y=1)}{P(X=1)}$ in Lemma 3, and therefore

$$\mathbb{E}_P[\phi] = \gamma P(X=1) = \gamma p_{+1}(x) = \alpha, \tag{43}$$

and

$$\beta_\phi^{x<x'}(\alpha) = 1 - \mathbb{E}_Q[\phi] = 1 - \gamma P(Y=1) = 1 - \gamma p_{+1}(x') = 1 - \frac{p_{+1}(x')}{p_{+1}(x)}\alpha. \tag{44}$$

When $\alpha \in [p_{+1}(x), 1]$, we set $h = \frac{P(Y=-1)}{P(X=-1)}$ in Lemma 3, and therefore

$$\mathbb{E}_P[\phi] = P(X=1) + \gamma P(X=-1) = p_{+1}(x) + \gamma p_{-1}(x) = \alpha, \tag{45}$$

and

$$\beta_\phi^{x<x'}(\alpha) = 1 - \mathbb{E}_Q[\phi] = 1 - P(Y=1) - \gamma P(Y=-1) = p_{-1}(x') - \gamma p_{-1}(x') = p_{-1}(x') - \frac{\alpha - p_{+1}(x)}{p_{-1}(x)}p_{-1}(x').$$

(46)

In summary, we have

$$\beta_\phi^{x<x'}(\alpha) = \begin{cases} 1 - \frac{p_{+1}(x')}{p_{+1}(x)}\alpha, & \text{for } \alpha \in [0, p_{+1}(x)], \\ \frac{p_{-1}(x')}{p_{-1}(x)} - \frac{p_{-1}(x')}{p_{-1}(x)}\alpha, & \text{for } \alpha \in [p_{+1}(x), 1]. \end{cases}$$

(47)

The infimum of $\beta_\phi^{x>x'}(\alpha)$ is attained when $p_{-1}(x') = p_{max}$ and $p_{-1}(x) = p_{min}$, while the infimum of $\beta_\phi^{x<x'}(\alpha)$ is attained when $p_{+1}(x') = p_{max}$ and $p_{+1}(x) = p_{min}$. As a result, we have

$$f^{NoisySign}(\alpha) = \begin{cases} 1 - \frac{p_{max}}{p_{min}}\alpha, & \text{for } \alpha \in [0, p_{min}], \\ \frac{p_{min}}{p_{max}} - \frac{p_{min}}{p_{max}}\alpha, & \text{for } \alpha \in [p_{min}, 1], \end{cases}$$

(48)

which completes the proof. $\qquad\square$

### A.2. Proof of Theorem 2

**Theorem 2.** *For any vectors $x$ and $x'$ with $||x||_2, ||x'||_2 \le C$, the tradeoff function $f^{GNS}(\alpha)$ for the corresponding hypothesis testing problem satisfies*

$$f^{GNS}(\alpha) \xrightarrow{d\to\infty} G_\mu(\alpha) = \Phi(\Phi^{-1}(1-\alpha) - \mu),$$

(49)

*in which $\mu$ is the limit of $\mu_d$ with*

$$\mu_d \le \frac{2C}{\sqrt{2\pi\Phi\left(\frac{C}{\sqrt{d}\sigma}\right)\Phi\left(-\frac{C}{\sqrt{d}\sigma}\right)}\sigma}.$$

(50)

*Proof.* In the vector case, we consider a higher dimensional hypothesis testing problem given by

$$H_0 : z \sim P = P_1 \times P_2 \times \cdots \times P_d,$$
$$H_1 : z \sim Q = Q_1 \times Q_2 \times \cdots \times Q_d.$$

(51)

Let $T_d(z) = \log\left(\frac{\prod_{i=1}^d q_i(z_i)}{\prod_{i=1}^d p_i(z_i)}\right)$ in which $p_i(\cdot)$ and $q_i(\cdot)$ denote the probability density functions of $P_i$ and $Q_i$, respectively, and $z = [z_1, z_2, ..., z_d]$ is the output of the mechanism. Similar to (Jang et al., 2024), the maximize of the privacy loss is given by $x' = -x = \frac{C}{\sqrt{d}} \cdot \mathbf{1}$.

According to the Neyman-Pearson lemma (Lehmann et al., 2005), the most powerful test at level $\alpha$ is a thresholding function of $T_d(z)$. Following the results in (Zheng et al., 2020) and (Dong et al., 2021), this is equivalent to applying some thresholding function $h(\alpha)$ to the normalized statistic $\frac{T_d(z) - \mathbb{E}_P[T_d(z)]}{\sqrt{\mathbb{V}\text{ar}_P[T_d(z)]}}$, i.e., $H_0$ is rejected if

$$\frac{T_d(z) - \mathbb{E}_P[T_d(z)]}{\sqrt{\mathbb{V}\text{ar}_P[T_d(z)]}} > h(\alpha).$$

(52)

Let $F_d(\cdot)$ be the CDF of $\frac{T_d(z) - \mathbb{E}_P[T_d(z)]}{\sqrt{\mathbb{V}\text{ar}_P[T_d(z)]}}$ when $z$ is drawn from $P$, i.e., $F_d(h) = \mathbb{P}_P\left(\frac{T_d(z) - \mathbb{E}_P[T_d(z)]}{\sqrt{\mathbb{V}\text{ar}_P[T_d(z)]}} \le h\right)$. By the Lyapunov central limit theorem, $\frac{T_d(z) - \mathbb{E}_P[T_d(z)]}{\sqrt{\mathbb{V}\text{ar}_P[T_d(z)]}}$ converges in distribution to the standard normal random variable (Billingsley, 2017). Similarly, the normalized statistic $\frac{T_d(z) - \mathbb{E}_Q[T_d(z)]}{\sqrt{\mathbb{V}\text{ar}_Q[T_d(z)]}}$ for $z \sim Q$ also converges in distribution to the standard normal random variable. Let $\tilde{F}_d(h) = \mathbb{P}_Q\left(\frac{T_d(z) - \mathbb{E}_Q[T_d(z)]}{\sqrt{\mathbb{V}\text{ar}_Q[T_d(z)]}} \le h\right)$.

Given the above results at hand, the type I error rate is given by

$$\mathbb{P}_P\left(\frac{T_d(z) - \mathbb{E}_P[T_d(z)]}{\sqrt{\mathbb{V}\text{ar}_P[T_d(z)]}} > h\right) = 1 - F_d(h) = \alpha,$$

(53)

and the type II error rate is given by

$$\mathbb{P}_Q \left( \frac{T_d(z) - \mathbb{E}_P[T_d(z)]}{\sqrt{\mathbb{V}\text{ar}_P[T_d(z)]}} \leq h \right) = \mathbb{P}_Q \left( \frac{T_d(z) - \mathbb{E}_Q[T_d(z)]}{\sqrt{\mathbb{V}\text{ar}_P[T_d(z)]}} \leq h - \frac{\mathbb{E}_Q[T_d(z)] - \mathbb{E}_P[T_d(z)]}{\sqrt{\mathbb{V}\text{ar}_P[T_d(z)]}} \right) = f(\alpha). \quad (54)$$

Since $f$ is symmetric, we can readily find that $\mathbb{V}\text{ar}_P[T_d(z)] = \mathbb{V}\text{ar}_Q[T_d(z)]$ (Dong et al., 2019; Zheng et al., 2020). As a result,

$$
\begin{aligned}
f(\alpha) &= \mathbb{P}_Q \left( \frac{T_d(z) - \mathbb{E}_Q[T_d(z)]}{\sqrt{\mathbb{V}\text{ar}_Q[T_d(z)]}} \leq h - \frac{\mathbb{E}_Q[T_d(z)] - \mathbb{E}_P[T_d(z)]}{\sqrt{\mathbb{V}\text{ar}_Q[T_d(z)]}} \right) \\
&= \tilde{F}_d \left( F_d^{-1}(1 - \alpha) - \frac{\mathbb{E}_Q[T_d(z)] - \mathbb{E}_P[T_d(z)]}{\sqrt{\mathbb{V}\text{ar}_Q[T_d(z)]}} \right).
\end{aligned}
\quad (55)
$$

Since both $F_d(\cdot)$ and $\tilde{F}_d(\cdot)$ converges in distribution to $\Phi(\cdot)$ when $d$ increases, we essentially have

$$f(\alpha) \to G_\mu(\alpha) = \Phi(\Phi^{-1}(1 - \alpha) - \mu), \quad (56)$$

in which $\mu$ is the limit of $\mu_d$ with

$$\mu_d = \frac{\mathbb{E}_Q[T_d(z)] - \mathbb{E}_P[T_d(z)]}{\sqrt{\mathbb{V}\text{ar}_Q[T_d(z)]}}. \quad (57)$$

In the following, we consider two input vectors $x$ and $x'$ such that $||x||_2 \leq C$ and $||x'||_2 \leq C$, and $z$ is the output of the NoisySign compressor with $z \sim P$ if the input vector is $x$ and $z \sim Q$ otherwise. In this case, we have

$$
\begin{aligned}
\mathbb{E}_P[T_d(z)] = \sum_i^d &\left[ \Phi\left(\frac{x_i}{\sigma}\right) \log\left(\Phi\left(\frac{x'_i}{\sigma}\right)\right) + \Phi\left(-\frac{x_i}{\sigma}\right) \log\left(\Phi\left(-\frac{x'_i}{\sigma}\right)\right) \right. \\
&\left. - \Phi\left(\frac{x_i}{\sigma}\right) \log\left(\Phi\left(\frac{x_i}{\sigma}\right)\right) - \Phi\left(-\frac{x_i}{\sigma}\right) \log\left(\Phi\left(-\frac{x_i}{\sigma}\right)\right) \right].
\end{aligned}
\quad (58)
$$

$$
\begin{aligned}
\mathbb{E}_Q[T_d(z)] = \sum_i^d &\left[ \Phi\left(\frac{x'_i}{\sigma}\right) \log\left(\Phi\left(\frac{x'_i}{\sigma}\right)\right) + \Phi\left(-\frac{x'_i}{\sigma}\right) \log\left(\Phi\left(-\frac{x'_i}{\sigma}\right)\right) \right. \\
&\left. - \Phi\left(\frac{x'_i}{\sigma}\right) \log\left(\Phi\left(\frac{x_i}{\sigma}\right)\right) - \Phi\left(-\frac{x'_i}{\sigma}\right) \log\left(\Phi\left(-\frac{x_i}{\sigma}\right)\right) \right].
\end{aligned}
\quad (59)
$$

Therefore,

$$\mathbb{E}_Q[T_d(z)] - \mathbb{E}_P[T_d(z)] = \sum_i^d \left[ \Phi\left(\frac{x'_i}{\sigma}\right) - \Phi\left(\frac{x_i}{\sigma}\right) \right] \log\left( \frac{\Phi\left(\frac{x'_i}{\sigma}\right) \Phi\left(-\frac{x_i}{\sigma}\right)}{\Phi\left(-\frac{x'_i}{\sigma}\right) \Phi\left(\frac{x_i}{\sigma}\right)} \right). \quad (60)$$

On the other hand, we have

$$\mathbb{V}\text{ar}_Q[T_d(z)] = \sum_i^d \left[ \Phi\left(\frac{x'_i}{\sigma}\right) \Phi\left(-\frac{x'_i}{\sigma}\right) \right] \log^2\left( \frac{\Phi\left(\frac{x'_i}{\sigma}\right) \Phi\left(-\frac{x_i}{\sigma}\right)}{\Phi\left(-\frac{x'_i}{\sigma}\right) \Phi\left(\frac{x_i}{\sigma}\right)} \right). \quad (61)$$

As a result,

$$\mu_d = \frac{\mathbb{E}_Q[T_d(z)] - \mathbb{E}_P[T_d(z)]}{\sqrt{\mathbb{V}\text{ar}_Q[T_d(z)]}} = \frac{\sum_i^d \left[ \Phi\left(\frac{x'_i}{\sigma}\right) - \Phi\left(\frac{x_i}{\sigma}\right) \right] \log\left( \frac{\Phi\left(\frac{x'_i}{\sigma}\right)\Phi\left(-\frac{x_i}{\sigma}\right)}{\Phi\left(-\frac{x'_i}{\sigma}\right)\Phi\left(\frac{x_i}{\sigma}\right)} \right)}{\sqrt{\sum_i^d \left[ \Phi\left(\frac{x'_i}{\sigma}\right) \Phi\left(-\frac{x'_i}{\sigma}\right) \right] \log^2\left( \frac{\Phi\left(\frac{x'_i}{\sigma}\right)\Phi\left(-\frac{x_i}{\sigma}\right)}{\Phi\left(-\frac{x'_i}{\sigma}\right)\Phi\left(\frac{x_i}{\sigma}\right)} \right)}}. \quad (62)$$

Plugging $x' = -x = \frac{C}{\sqrt{d}} \cdot \mathbf{1}$ into (62) yields

$$\mu_d = \frac{\mathbb{E}_Q\left[T_d(\boldsymbol{z})\right] - \mathbb{E}_P\left[T_d(\boldsymbol{z})\right]}{\sqrt{\mathbb{Var}_Q\left[T_d(\boldsymbol{z})\right]}} = \frac{\left[\Phi\left(\frac{C}{\sqrt{d}\sigma}\right) - \Phi\left(-\frac{C}{\sqrt{d}\sigma}\right)\right]\sqrt{d}}{\sqrt{\Phi\left(\frac{C}{\sqrt{d}\sigma}\right)\Phi\left(-\frac{C}{\sqrt{d}\sigma}\right)}}. \tag{63}$$

The mean value theorem (Rudin et al., 1964) implies that

$$\left|\Phi\left(\frac{\boldsymbol{x}_i'}{\sigma}\right) - \Phi\left(\frac{\boldsymbol{x}_i}{\sigma}\right)\right| \leq \frac{1}{\sigma\sqrt{2\pi}}\left|\boldsymbol{x}_i' - \boldsymbol{x}_i\right|. \tag{64}$$

Therefore, we conclude that

$$\mu_d \leq \frac{2C}{\sqrt{2\pi\Phi\left(\frac{C}{\sqrt{d}\sigma}\right)\Phi\left(-\frac{C}{\sqrt{d}\sigma}\right)}\sigma}, \tag{65}$$

which completes the proof.

$$\square$$

### A.3. Proof of Theorem 3

**Theorem 3** (**Convergence of NoisySign SGD with Scheme I**). *Suppose Assumptions 1-4 are satisfied, and the learning rate is set as* $\eta = \frac{1}{\sqrt{TLd}}$. *Then by running Algorithm 2 for $T$ iterations, we have*

$$\frac{1}{T}\sum_{t=1}^{T}\left\|\left|\nabla F(\boldsymbol{w}^{(t)})\right|\right\|_2^2 \leq \frac{(F(\boldsymbol{w}^{(0)}) - F^*)\sqrt{Ld}\sqrt{2\pi}\sigma_{DP}}{\sqrt{T}} + \frac{\sqrt{Ld}\sqrt{2\pi}\sigma_{DP}}{2\sqrt{T}} + \mathcal{O}\left(\frac{c^6}{\sigma_{DP}^4}\right). \tag{66}$$

Before proving Theorem 3, we first show the following lemma.

**Lemma 4.** *For any $x \in [-c, c]$ and $\sigma_{DP} > c$, we have*

$$\mathbb{E}[sign\left(x + \mathcal{N}(0, \sigma_{DP})\right)] = \frac{2}{\sigma_{DP}\sqrt{2\pi}}x + \Delta, \tag{67}$$

*and*

$$\mathbb{E}[|sign\left(x + \mathcal{N}(0, \sigma_{DP})\right) - \mathbb{E}[sign\left(x + \mathcal{N}(0, \sigma_{DP})\right)]|^2] \leq 1, \tag{68}$$

*in which $\Delta = \mathcal{O}\left(\frac{c^3}{\sigma_{DP}^3}\right)$.*

*Proof.*

$$\begin{aligned}
&\mathbb{E}[sign\left(x + \mathcal{N}(0, \sigma_{DP})\right)] \\
&= \Phi\left(\frac{x}{\sigma_{DP}}\right) - \Phi\left(-\frac{x}{\sigma_{DP}}\right) \\
&= \frac{1}{\sqrt{2\pi}}\int_{-\infty}^{\frac{x}{\sigma_{DP}}} e^{-\frac{t^2}{2}}\,dt - \frac{1}{\sqrt{2\pi}}\int_{-\infty}^{-\frac{x}{\sigma_{DP}}} e^{-\frac{t^2}{2}}\,dt \\
&= \frac{1}{\sqrt{2\pi}}\int_{-\frac{x}{\sigma_{DP}}}^{\frac{x}{\sigma_{DP}}} e^{-\frac{t^2}{2}}\,dt \\
&= \frac{1}{\sqrt{2\pi}}\int_{-\frac{x}{\sigma_{DP}}}^{\frac{x}{\sigma_{DP}}}\sum_{k=0}^{\infty}(-1)^k\frac{x^{2k}}{2^k k!}\,dt \\
&= \frac{1}{\sqrt{2\pi}}\int_{-\frac{x}{\sigma_{DP}}}^{\frac{x}{\sigma_{DP}}} 1 + \sum_{k=1}^{\infty}(-1)^k\frac{x^{2k}}{2^k k!}\,dt \\
&= \frac{2}{\sqrt{2\pi}\sigma_{DP}}x + \frac{1}{\sqrt{2\pi}}\left[\sum_{k=1}^{\infty}(-1)^k\frac{x^{2k+1}}{2^k k!(2k+1)}\right]_{-\frac{x}{\sigma_{DP}}}^{\frac{x}{\sigma_{DP}}} \\
&= \frac{2}{\sigma_{DP}\sqrt{2\pi}}x + \mathcal{O}\left(\frac{c^3}{\sigma_{DP}^3}\right),
\end{aligned} \tag{69}$$

in which we utilize the Taylor expansion of $e^{-\frac{t^2}{2}}$.

Moreover,

$$\mathbb{E}[|sign\left(x + \mathcal{N}(0, \sigma_{DP})\right) - \mathbb{E}[sign\left(x + \mathcal{N}(0, \sigma_{DP})\right)]|^2]$$

$$= \Phi\left(\frac{x}{\sigma_{DP}}\right)|1 - \mathbb{E}[sign\left(x + \mathcal{N}(0, \sigma_{DP})\right)]|^2 + \Phi\left(-\frac{x}{\sigma_{DP}}\right)|1 + \mathbb{E}[sign\left(x + \mathcal{N}(0, \sigma_{DP})\right)]|^2$$

$$= \left[\Phi\left(\frac{x}{\sigma_{DP}}\right) + \Phi\left(-\frac{x}{\sigma_{DP}}\right)\right]\left[1 + (\mathbb{E}[sign\left(x + \mathcal{N}(0, \sigma_{DP})\right)])^2\right] \tag{70}$$

$$- 2\left[\Phi\left(\frac{x}{\sigma_{DP}}\right) - \Phi\left(-\frac{x}{\sigma_{DP}}\right)\right]\mathbb{E}[sign\left(x + \mathcal{N}(0, \sigma_{DP})\right)]$$

$$= 1 - (\mathbb{E}[sign\left(x + \mathcal{N}(0, \sigma_{DP})\right)])^2 \leq 1,$$

which completes the proof. $\qquad\square$

Given Lemma 4, we are ready to prove Theorem 3.

*Proof.* According to Assumption 2, we have

$$F(\boldsymbol{w}^{(t+1)}) - F(\boldsymbol{w}^{(t)})$$

$$\leq \langle \nabla F(\boldsymbol{w}^{(t)}), \boldsymbol{w}^{(t+1)} - \boldsymbol{w}^{(t)}\rangle + \frac{L}{2}||\boldsymbol{w}^{(t+1)} - \boldsymbol{w}^{(t)}||_2^2$$

$$= -\eta\left\langle \nabla F(\boldsymbol{w}^{(t)}), \frac{1}{M}\sum_{m\in\mathcal{H}} sign(\boldsymbol{g}_m^{(t)} + \boldsymbol{n}_m)\right\rangle + \frac{L}{2}\left|\left|\eta\frac{1}{M}\sum_{m\in\mathcal{H}} sign(\boldsymbol{g}_m^{(t)} + \boldsymbol{n}_m)\right|\right|^2 \tag{71}$$

$$\leq -\eta\left\langle \nabla F(\boldsymbol{w}^{(t)}), \frac{1}{M}\sum_{m\in\mathcal{H}} sign(\boldsymbol{g}_m^{(t)} + \boldsymbol{n}_m)\right\rangle + \frac{Ld\eta^2}{2}$$

Taking expectations on both sides yields

$$\mathbb{E}[F(\boldsymbol{w}^{(t+1)}) - F(\boldsymbol{w}^{(t)})]$$

$$\leq -\eta\mathbb{E}\left[\left\langle \nabla F(\boldsymbol{w}^{(t)}), \frac{1}{M}\sum_{m\in\mathcal{H}} sign(\boldsymbol{g}_m^{(t)} + \boldsymbol{n}_m)\right\rangle\right] + \frac{Ld\eta^2}{2}$$

$$= -\eta\mathbb{E}\left[\left\langle \nabla F(\boldsymbol{w}^{(t)}), \frac{2}{\sigma_{DP}\sqrt{2\pi}}\frac{1}{M}\sum_{m\in\mathcal{H}} \boldsymbol{g}_m^{(t)} + \frac{1}{M}\sum_{m\in\mathcal{H}} \boldsymbol{\Delta}_m\right\rangle\right] + \frac{Ld\eta^2}{2}$$

$$= -\eta\frac{2}{\sigma_{DP}\sqrt{2\pi}}\left|\left|\nabla F(\boldsymbol{w}^{(t)})\right|\right|_2^2 - \eta\mathbb{E}\left[\left\langle \nabla F(\boldsymbol{w}^{(t)}), \frac{1}{M}\sum_{m\in\mathcal{H}} \boldsymbol{\Delta}_m\right\rangle\right] + \frac{Ld\eta^2}{2} \tag{72}$$

$$= -\eta\frac{2}{\sigma_{DP}\sqrt{2\pi}}\left|\left|\nabla F(\boldsymbol{w}^{(t)})\right|\right|_2^2 + \frac{\eta}{2}\mathbb{E}\left[\frac{2}{\sigma_{DP}\sqrt{2\pi}}\left|\left|\nabla F(\boldsymbol{w}^{(t)})\right|\right|_2^2 + \frac{\sqrt{2\pi}\sigma_{DP}}{2}\left|\left|\frac{1}{M}\sum_{m\in\mathcal{H}} \boldsymbol{\Delta}_m\right|\right|_2^2\right] + \frac{Ld\eta^2}{2}$$

$$= -\frac{\eta}{\sigma_{DP}\sqrt{2\pi}}\left|\left|\nabla F(\boldsymbol{w}^{(t)})\right|\right|_2^2 + \eta\mathcal{O}\left(\frac{c^6}{\sigma_{DP}^5}\right) + \frac{Ld\eta^2}{2},$$

in which $\boldsymbol{\Delta}_m = [\boldsymbol{\Delta}_{m,1}, \boldsymbol{\Delta}_{m,2}, ..., \boldsymbol{\Delta}_{m,d}]$ with $\boldsymbol{\Delta}_{m,i} = \mathcal{O}\left(\frac{c^3}{\sigma_{DP}^3}\right) \forall i$, and we utilize the fact that $- < a, b > \leq \frac{1}{2}||a||_2^2 + \frac{1}{2}||b||_2^2$. Adjusting the above inequality and averaging both sides over $t = 1, 2, \cdots, T$, we can obtain

$$\frac{1}{T}\sum_{t=1}^{T}\left|\left|\nabla F(\boldsymbol{w}^{(t)})\right|\right|_2^2 \leq \frac{\mathbb{E}[F(\boldsymbol{w}^{(0)}) - F(\boldsymbol{w}^{(t+1)})]\sigma_{DP}\sqrt{2\pi}}{T\eta} + \frac{Ld\sqrt{2\pi}\eta\sigma_{DP}}{2} + \mathcal{O}\left(\frac{c^6}{\sigma_{DP}^4}\right). \tag{73}$$

Letting $\eta = \frac{1}{\sqrt{LTd}}$ gives

$$
\begin{aligned}
&\frac{1}{T} \sum_{t=1}^{T} \left\| \nabla F(\boldsymbol{w}^{(t)}) \right\|_2^2 \\
&\leq \frac{\mathbb{E}[F(\boldsymbol{w}^{(0)}) - F(\boldsymbol{w}^{(t+1)})] \sqrt{Ld} \sqrt{2\pi} \sigma_{DP}}{\sqrt{T}} + \frac{\sqrt{Ld} \sqrt{2\pi} \sigma_{DP}}{2\sqrt{T}} + \mathcal{O}\left( \frac{c^6}{\sigma_{DP}^4} \right) \\
&\leq \frac{(F(\boldsymbol{w}^{(0)}) - F^*) \sqrt{Ld} \sqrt{2\pi} \sigma_{DP}}{\sqrt{T}} + \frac{\sqrt{Ld} \sqrt{2\pi} \sigma_{DP}}{2\sqrt{T}} + \mathcal{O}\left( \frac{c^6}{\sigma_{DP}^4} \right).
\end{aligned}
\tag{74}
$$

which completes the proof. $\qquad\qquad\square$

## B. Proof of Theorem 4

**Theorem 4** (**Convergence of NoisySign SGD with Scheme II**). *Suppose Assumptions 1-4 are satisfied, and the learning rate is set as $\eta = \frac{1}{\sqrt{TLd}}$. Then by running Algorithm 2 for $T$ iterations, we have*

$$
\begin{aligned}
\frac{1}{T} \sum_{t=1}^{T} \|\nabla F(\boldsymbol{w}^{(t)})\|_1 &\leq \frac{(F(\boldsymbol{w}^{(0)}) - F^*) \sqrt{Ld}}{\sqrt{T}} + \frac{\sqrt{Ld}}{2\sqrt{T}} + \left[ \frac{\sigma_{DP} d \sqrt{2\pi}}{\sqrt{M}} + \frac{2\|\bar{\boldsymbol{\sigma}}_L\|_1}{\sqrt{M}} + \mathcal{O}\left( \frac{c^3}{\sigma_{DP}^2} \right) \right] \\
&\leq \mathcal{O}\left( \frac{1}{\sqrt{T}} \right) + \mathcal{O}\left( \frac{\sigma_{DP}}{\sqrt{M}} + \frac{\|\bar{\boldsymbol{\sigma}}_L\|_1}{\sqrt{M}} \right) + \mathcal{O}\left( \frac{c^3}{\sigma_{DP}^2} \right).
\end{aligned}
\tag{75}
$$

*Proof.* According to Assumption 2, we have

$$
\begin{aligned}
&F(\boldsymbol{w}^{(t+1)}) - F(\boldsymbol{w}^{(t)}) \\
&\leq \langle \nabla F(\boldsymbol{w}^{(t)}), \boldsymbol{w}^{(t+1)} - \boldsymbol{w}^{(t)} \rangle + \frac{L}{2} \|\boldsymbol{w}^{(t+1)} - \boldsymbol{w}^{(t)}\|_2^2 \\
&= -\eta \left\langle \nabla F(\boldsymbol{w}^{(t)}), sign\left( \frac{1}{M} \sum_{m \in \mathcal{H}} sign(\boldsymbol{g}_m^{(t)} + \boldsymbol{n}_m) \right) \right\rangle \\
&\quad + \frac{L}{2} \left\| \eta \, sign\left( \frac{1}{M} \sum_{m \in \mathcal{H}} sign(\boldsymbol{g}_m^{(t)} + \boldsymbol{n}_m) \right) \right\|^2 \\
&\leq -\eta \left\langle \nabla F(\boldsymbol{w}^{(t)}), sign\left( \frac{1}{M} \sum_{m \in \mathcal{H}} sign(\boldsymbol{g}_m^{(t)} + \boldsymbol{n}_m) \right) \right\rangle + \frac{Ld\eta^2}{2} \\
&= -\eta \|\nabla F(\boldsymbol{w}^{(t)})\|_1 + \frac{Ld\eta^2}{2} + 2\eta \sum_{i=1}^{d} |\nabla F(\boldsymbol{w}^{(t)})_i| \times \mathbb{1}_{sign(\frac{1}{M} \sum_{m \in \mathcal{H}} sign(\boldsymbol{g}_{m,i}^{(t)} + \boldsymbol{n}_m)) \neq sign(\nabla F(\boldsymbol{w}^{(t)})_i)},
\end{aligned}
\tag{76}
$$

where $\nabla F(\boldsymbol{w}^{(t)})_i$ is the $i$-th entry of the vector $\nabla F(\boldsymbol{w}^{(t)})$ and $\eta$ is the learning rate. Taking expectations on both sides yields

$$
\begin{aligned}
&\mathbb{E}[F(\boldsymbol{w}^{(t+1)}) - F(\boldsymbol{w}^{(t)})] \\
&\leq -\eta \|\nabla F(\boldsymbol{w}^{(t)})\|_1 + \frac{Ld\eta^2}{2} \\
&\quad + 2\eta \sum_{i=1}^{d} \mathbb{E}\left[ |\nabla F(\boldsymbol{w}^{(t)})_i| P\left( sign\left( \frac{1}{M} \sum_{m \in \mathcal{H}} sign(\boldsymbol{g}_{m,i}^{(t)} + \boldsymbol{n}_m) \right) \neq sign(\nabla F(\boldsymbol{w}^{(t)})_i) \right) \right].
\end{aligned}
\tag{77}
$$

In addition,

$$
P\left(sign\left(\frac{1}{M}\sum_{m\in\mathcal{H}}sign(\boldsymbol{g}_{m,i}^{(t)}+\boldsymbol{n}_m)\right)\neq sign(\nabla F(\boldsymbol{w}^{(t)})_i)\right)
$$

$$
= P\left(sign\left(\frac{1}{M}\sum_{m\in\mathcal{H}}sign(\boldsymbol{g}_{m,i}^{(t)}+\boldsymbol{n}_m)\right)\neq sign\left(\frac{2}{\sigma_{DP}\sqrt{2\pi}}\nabla F(\boldsymbol{w}^{(t)})_i\right)\right)
$$

$$
\leq P\left(\left|\frac{1}{M}\sum_{m\in\mathcal{H}}sign(\boldsymbol{g}_{m,i}^{(t)}+\boldsymbol{n}_m)-\frac{2}{\sigma_{DP}\sqrt{2\pi}}\nabla F(\boldsymbol{w}^{(t)})_i\right|\geq\left|\frac{2}{\sigma_{DP}\sqrt{2\pi}}\nabla F(\boldsymbol{w}^{(t)})_i\right|\right)
$$

$$
\leq\frac{\mathbb{E}\left[\left|\frac{1}{M}\sum_{m\in\mathcal{H}}sign(\boldsymbol{g}_{m,i}^{(t)}+\boldsymbol{n}_m)-\frac{2}{\sigma_{DP}\sqrt{2\pi}}\nabla F(\boldsymbol{w}^{(t)})_i\right|\right]}{\frac{2}{\sigma_{DP}\sqrt{2\pi}}|\nabla F(\boldsymbol{w}^{(t)})_i|}
\tag{78}
$$

$$
\leq\frac{\mathbb{E}\left[\left|\frac{1}{M}\sum_{m\in\mathcal{H}}sign(\boldsymbol{g}_{m,i}^{(t)}+\boldsymbol{n}_m)-\frac{1}{M}\sum_{m\in\mathcal{H}}\mathbb{E}\left[sign(\boldsymbol{g}_{m,i}^{(t)}+\boldsymbol{n}_m)\right]\right|\right]}{\frac{2}{\sigma_{DP}\sqrt{2\pi}}|\nabla F(\boldsymbol{w}^{(t)})_i|}
$$

$$
+\frac{\mathbb{E}\left[\left|\frac{1}{M}\sum_{m\in\mathcal{H}}\mathbb{E}\left[sign(\boldsymbol{g}_{m,i}^{(t)}+\boldsymbol{n}_m)\right]-\frac{1}{M}\sum_{m\in\mathcal{H}}\frac{2}{\sigma_{DP}\sqrt{2\pi}}\boldsymbol{g}_{m,i}^{(t)}\right|\right]}{\frac{2}{\sigma_{DP}\sqrt{2\pi}}|\nabla F(\boldsymbol{w}^{(t)})_i|}
$$

$$
+\frac{\mathbb{E}\left[\left|\frac{1}{M}\sum_{m\in\mathcal{H}}\frac{2}{\sigma_{DP}\sqrt{2\pi}}\boldsymbol{g}_{m,i}^{(t)}-\frac{2}{\sigma_{DP}\sqrt{2\pi}}\nabla F(\boldsymbol{w}^{(t)})_i\right|\right]}{\frac{2}{\sigma_{DP}\sqrt{2\pi}}|\nabla F(\boldsymbol{w}^{(t)})_i|}.
$$

For each term above, we have

$$
\sum_{i=1}^{d}\mathbb{E}\left[\left|\frac{1}{M}\sum_{m\in\mathcal{H}}sign(\boldsymbol{g}_{m,i}^{(t)}+\boldsymbol{n}_m)-\frac{1}{M}\sum_{m\in\mathcal{H}}\mathbb{E}\left[sign(\boldsymbol{g}_{m,i}^{(t)}+\boldsymbol{n}_m)\right]\right|\right]
$$

$$
=\sum_{i=1}^{d}\sqrt{\left[\mathbb{E}\left[\left|\frac{1}{M}\sum_{m\in\mathcal{H}}sign(\boldsymbol{g}_{m,i}^{(t)}+\boldsymbol{n}_m)-\frac{1}{M}\sum_{m\in\mathcal{H}}\mathbb{E}\left[sign(\boldsymbol{g}_{m,i}^{(t)}+\boldsymbol{n}_m)\right]\right|\right]\right]^2}
\tag{79}
$$

$$
\leq\sum_{i=1}^{d}\sqrt{\mathbb{E}\left[\left|\frac{1}{M}\sum_{m\in\mathcal{H}}sign(\boldsymbol{g}_{m,i}^{(t)}+\boldsymbol{n}_m)-\frac{1}{M}\sum_{m\in\mathcal{H}}\mathbb{E}\left[sign(\boldsymbol{g}_{m,i}^{(t)}+\boldsymbol{n}_m)\right]\right|^2\right]}
$$

$$
\leq\sum_{i=1}^{d}\sqrt{\frac{1}{M^2}\sum_{m\in\mathcal{H}}\mathbb{E}\left|sign(\boldsymbol{g}_{m,i}^{(t)}+\boldsymbol{n}_m)-\mathbb{E}\left[sign(\boldsymbol{g}_{m,i}^{(t)}+\boldsymbol{n}_m)\right]\right|^2}\leq\frac{d}{\sqrt{M}},
$$

$$\sum_{i=1}^{d} \mathbb{E}\left[\left|\frac{2}{\sigma_{DP}\sqrt{2\pi}}\nabla F(\boldsymbol{w}^{(t)})_i - \frac{2}{\sigma_{DP}\sqrt{2\pi}}\frac{1}{M}\sum_{m\in\mathcal{H}}\boldsymbol{g}_{m,i}^{(t)}\right|\right]$$

$$= \sum_{i=1}^{d}\sqrt{\left[\mathbb{E}\left[\left|\frac{2}{\sigma_{DP}\sqrt{2\pi}}\nabla F(\boldsymbol{w}^{(t)})_i - \frac{1}{M}\sum_{m\in\mathcal{H}}\frac{2}{\sigma_{DP}\sqrt{2\pi}}\boldsymbol{g}_{m,i}^{(t)}\right|\right]\right]^2}$$

$$\leq \sum_{i=1}^{d}\sqrt{\mathbb{E}\left[\left|\frac{2}{\sigma_{DP}\sqrt{2\pi}}\nabla F(\boldsymbol{w}^{(t)})_i - \frac{1}{M}\sum_{m\in\mathcal{H}}\frac{2}{\sigma_{DP}\sqrt{2\pi}}\boldsymbol{g}_{m,i}^{(t)}\right|^2\right]}$$

$$= \sum_{i=1}^{d}\sqrt{\mathbb{E}\left[\left|\frac{1}{M}\sum_{m\in\mathcal{H}}\frac{2}{\sigma_{DP}\sqrt{2\pi}}\nabla F_m(\boldsymbol{w}^{(t)})_i - \frac{1}{M}\sum_{m\in\mathcal{H}}\frac{2}{\sigma_{DP}\sqrt{2\pi}}\boldsymbol{g}_{m,i}^{(t)}\right|^2\right]}$$

$$= \sum_{i=1}^{d}\sqrt{\frac{1}{M^2}\sum_{m\in\mathcal{H}}\mathbb{E}[|\frac{2}{\sigma_{DP}\sqrt{2\pi}}\nabla F_m(\boldsymbol{w}^{(t)})_i - \frac{2}{\sigma_{DP}\sqrt{2\pi}}\boldsymbol{g}_{m,i}^{(t)}|^2]}$$

$$\leq \sum_{i=1}^{d}\sqrt{\frac{\sigma_{L,i}^2}{M}}\frac{2}{\sigma_{DP}\sqrt{2\pi}} = \frac{||2\bar{\boldsymbol{\sigma}}_L||_1}{\sigma_{DP}\sqrt{2\pi M}}, \tag{80}$$

and

$$\mathbb{E}\left[\left|\frac{1}{M}\sum_{m\in\mathcal{H}}\mathbb{E}\left[sign(\boldsymbol{g}_{m,i}^{(t)}+\boldsymbol{n}_m)\right] - \frac{1}{M}\sum_{m\in\mathcal{H}}\frac{2}{\sigma_{DP}\sqrt{2\pi}}\boldsymbol{g}_{m,i}^{(t)}\right|\right] \leq \mathcal{O}\left(\frac{c^3}{\sigma_{DP}^3}\right). \tag{81}$$

Plugging (78),(79),(80), and (81) into (77) yields

$$\mathbb{E}[F(\boldsymbol{w}^{(t+1)}) - F(\boldsymbol{w}^{(t)})]$$

$$\leq -\eta||\nabla F(\boldsymbol{w}^{(t)})||_1 + \frac{Ld\eta^2}{2} + 2\eta\left[\frac{d}{\sqrt{M}} + \frac{||2\bar{\boldsymbol{\sigma}}_L||_1}{\sigma_{DP}\sqrt{2\pi M}} + \mathcal{O}\left(\frac{c^3}{\sigma_{DP}^3}\right)\right]\frac{\sigma_{DP}\sqrt{2\pi}}{2}$$

$$= -\eta||\nabla F(\boldsymbol{w}^{(t)})||_1 + \frac{Ld\eta^2}{2} + \eta\left[\frac{\sigma_{DP}d\sqrt{2\pi}}{\sqrt{M}} + \frac{2||\bar{\boldsymbol{\sigma}}_L||_1}{\sqrt{M}} + \mathcal{O}\left(\frac{c^3}{\sigma_{DP}^2}\right)\right]. \tag{82}$$

Adjusting the above inequality and averaging both sides over $t = 1, 2, \cdots, T$, we can obtain

$$\frac{1}{T}\sum_{t=1}^{T}\eta||\nabla F(\boldsymbol{w}^{(t)})||_1$$

$$\leq \frac{\mathbb{E}[F(\boldsymbol{w}^{(0)}) - F(\boldsymbol{w}^{(t+1)})]}{T} + \frac{Ld\eta^2}{2} + \eta\left[\frac{\sigma_{DP}d\sqrt{2\pi}}{\sqrt{M}} + \frac{2||\bar{\boldsymbol{\sigma}}_L||_1}{\sqrt{M}} + \mathcal{O}\left(\frac{c^3}{\sigma_{DP}^2}\right)\right]. \tag{83}$$

Letting $\eta = \frac{1}{\sqrt{LTd}}$ and dividing both sides by $\eta$ gives

$$\frac{1}{T}\sum_{t=1}^{T}||\nabla F(\boldsymbol{w}^{(t)})||_1$$

$$\leq \frac{\mathbb{E}[F(\boldsymbol{w}^{(0)}) - F(\boldsymbol{w}^{(t+1)})]\sqrt{Ld}}{\sqrt{T}} + \frac{\sqrt{Ld}}{2\sqrt{T}} + \left[\frac{\sigma_{DP}d\sqrt{2\pi}}{\sqrt{M}} + \frac{2||\bar{\boldsymbol{\sigma}}_L||_1}{\sqrt{M}} + \mathcal{O}\left(\frac{c^3}{\sigma_{DP}^2}\right)\right]$$

$$\leq \frac{(F(\boldsymbol{w}^{(0)}) - F^*)\sqrt{Ld}}{\sqrt{T}} + \frac{\sqrt{Ld}}{2\sqrt{T}} + \left[\frac{\sigma_{DP}d\sqrt{2\pi}}{\sqrt{M}} + \frac{2||\bar{\boldsymbol{\sigma}}_L||_1}{\sqrt{M}} + \mathcal{O}\left(\frac{c^3}{\sigma_{DP}^2}\right)\right]. \tag{84}$$

which completes the proof. $\qquad\square$

## C. Details of the Implementation and Additional Results

Our experiments are mainly implemented using Python 3.8 with packages Numpy 1.19.2 and Pytorch 1.10.1.

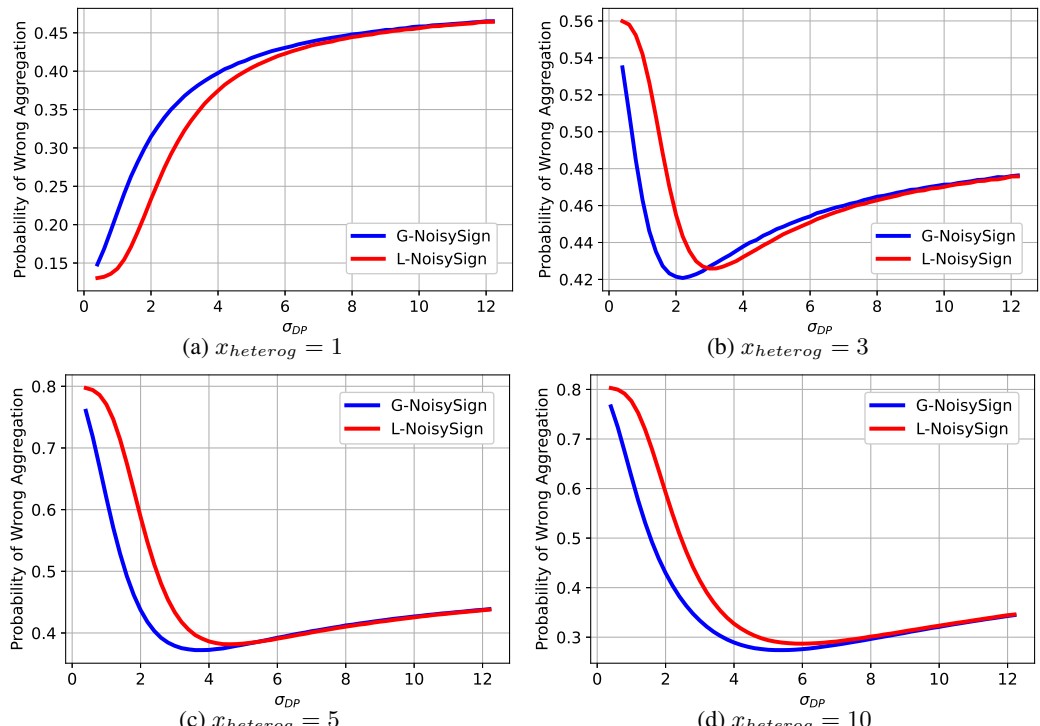

*Figure 4.* The comparison of the probability of wrong aggregation between G-NoisySign and L-NoisySign with $c = 10$. A set of 10 workers is considered, with $x_m = \mathcal{N}(-1, 1)$ for $m < 7$ and $x_m = \mathcal{N}(x_{heterog}, 1)$ otherwise. For the left and right figures in the first row, $x_{heterog} = 1$ and $x_{heterog} = 3$, respectively. For the left and right figures in the second row, $x_{heterog} = 5$ and $x_{heterog} = 10$, respectively.

### C.1. Dataset and Pre-processing

We perform experiments on the standard Fashion-MNIST (Xiao et al., 2017) and CIFAR-10 (Krizhevsky et al., 2009) datasets. The Fashion-MNIST dataset consists of 60,000 training samples and 10,000 testing samples. Each sample is a 28×28 size gray-level image. We normalize the data by dividing it by the max RGB value (i.e., 255.0). The CIFAR-10 dataset contains 50,000 training samples and 10,000 testing samples. Each sample is a 32×32 color image. The data are normalized with a zero-centered mean.

### C.2. Additional Results

#### C.2.1. ADDITIONAL RESULTS FOR SECTION 4

We conduct additional experiments to examine the impact of data heterogeneity. In Figure 4, a set of 10 workers is considered, with $x_m = \mathcal{N}(-1, 1)$ for $m < 7$ and $x_m = \mathcal{N}(x_{heterog}, 1)$ otherwise. It can be observed that when the data distribution is less heterogeneous (i.e., $x_{heterog} = 1$), L-NoisySign outperforms G-NoisySign since the vanilla SIGNSGD converges well and Gaussian noise tends to result in a larger variance than Logistic noise with the same privacy guarantees. When the data distribution becomes more heterogeneous, G-NoisySign outperforms L-NoisySign for an appropriate $\sigma_{DP}$. The crossover happens when $\sigma_{DP}$ is larger for more severe data heterogeneity. This validates that G-NoisySign may be more suitable for heterogeneous cases. We further consider a set of 10 workers, with $x_m = -1$ for $m < 7$ and $x_m = x_{heterog}$ otherwise, and the results are presented in Figure 5, which agree with those in Figure 4.

#### C.2.2. ADDITIONAL RESULTS FOR SECTION 6

Figure 6 presents the convergence of the algorithms with respect to the communication rounds. Note that compared to Gaussian noise (i.e., DP-SGD), G-NoisySign and L-NoisySign reduce the communication overhead from the workers to the server by a factor of 32, while the majority vote variants further enjoy improvement in communication efficiency from the server to the workers.

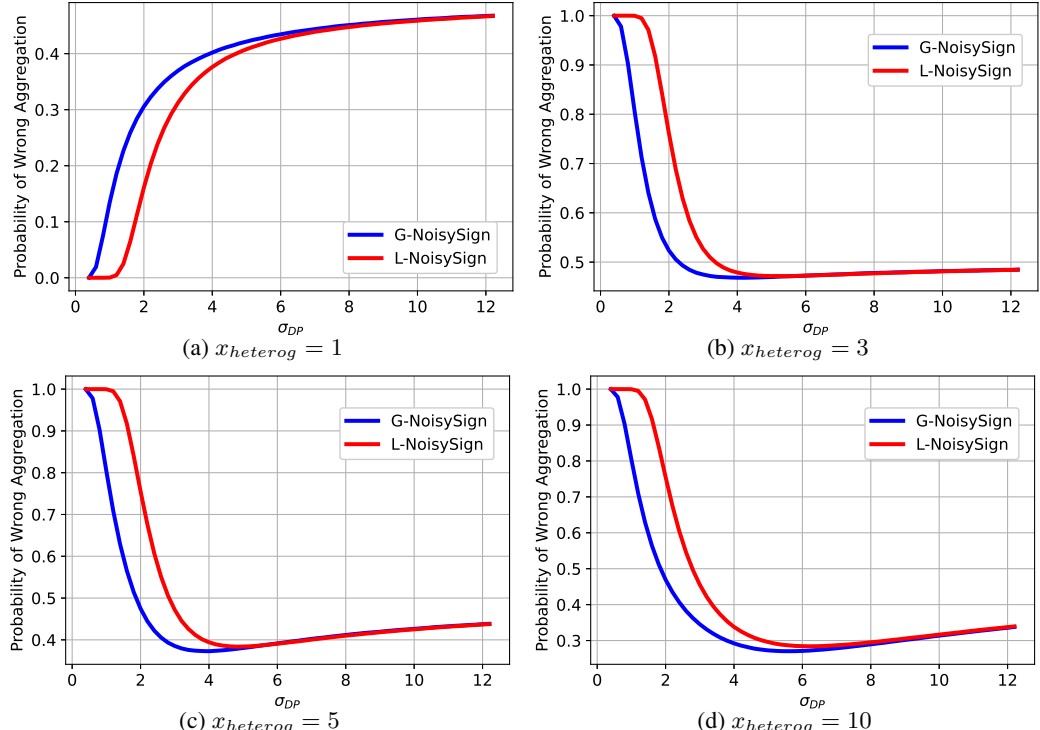

*Figure 5.* The comparison of the probability of wrong aggregation between G-NoisySign and L-NoisySign with $c = 10$. A set of 10 workers is considered, with $x_m = -1$ for $m < 7$ and $x_m = x_{heterog}$ otherwise. The results agree with those in Figure 4.

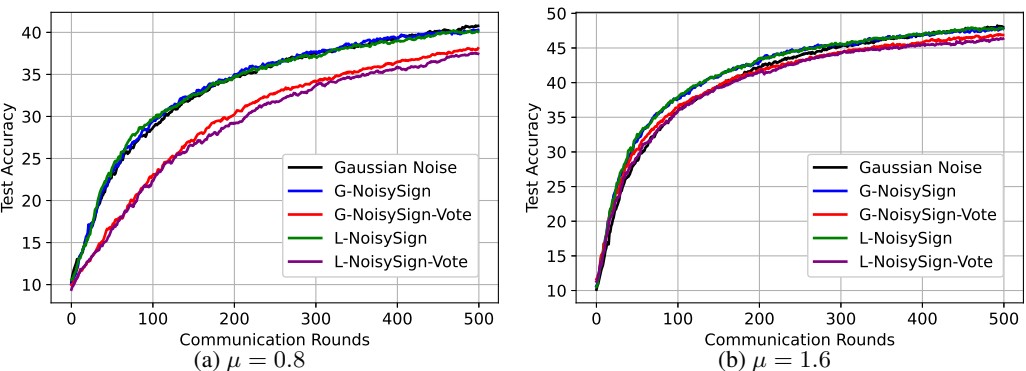

*Figure 6.* The convergence curve of the algorithms on CIFAR-10 with $\mu \in \{0.8, 1.6\}$.

We note that the composition property in Lemma 2 suggests that, for the same overall privacy guarantee, utilizing the privacy amplification effect of $sign(\cdot)$ allows for $\frac{\pi}{2} \approx 1.5\times$ training steps with the same Gaussian noise variance per step. Therefore, in Table 3, we add two more baselines: (1) G-NoisySign without utilizing the privacy amplification analyses (i.e., adding Gaussian noise with the same variance as DP-SGD); (2) G-NoisySign that runs 320 rounds (slightly more than $500 * 2/\pi$ rounds) instead of 500 rounds. It can be observed that G-NoisySign and G-NoisySign-Vote outperform the two additional baselines in all the examined privacy budgets, which validates that the privacy amplification indeed leads to improvement in test accuracy.

In Table 4-5, we present the results for a less heterogeneous data distribution, with $\alpha = 100$ and the remaining settings the same as Table 1-2. In Table 6, we perform experiments on Fashion-MNIST with 5 workers selected in each communication round. It can be observed that G-NoisySign and L-NoisySign achieve performance comparable to the Gaussian mechanism in all the examined scenarios.

*Table 3.* Fashion-MNIST Test Accuracy for Varying Privacy Requirements Per Communication Round ($\alpha = 0.1$)

| $\mu$ | 0.04 | 0.08 | 0.4 | 0.8 | 1.6 |
|---|---|---|---|---|---|
| GAUSSIAN NOISE | $43.78 \pm 1.98\%$ | $58.69 \pm 1.48\%$ | $73.48 \pm 0.49\%$ | $77.17 \pm 0.51\%$ | $79.90 \pm 0.25\%$ |
| G-NOISYSIGN | $45.24 \pm 2.51\%$ | $58.46 \pm 2.48\%$ | $73.78 \pm 0.57\%$ | $77.20 \pm 0.29\%$ | $79.57 \pm 0.57\%$ |
| G-NOISYSIGN (320 ROUND) | $40.94 \pm 4.49\%$ | $54.49 \pm 2.68\%$ | $72.71 \pm 0.68\%$ | $76.15 \pm 0.46\%$ | $78.70 \pm 0.65\%$ |
| G-NOISYSIGN W.O. PRIVACY AMPLIFICATION | $37.36 \pm 4.07\%$ | $52.06 \pm 2.54\%$ | $72.83 \pm 0.62\%$ | $76.49 \pm 0.43\%$ | $79.11 \pm 0.45\%$ |
| G-NOISYSIGN-VOTE | $42.18 \pm 4.03\%$ | $56.59 \pm 1.91\%$ | $73.23 \pm 0.66\%$ | $76.47 \pm 0.61\%$ | $79.27 \pm 0.37\%$ |
| G-NOISYSIGN-VOTE (320 ROUND) | $35.79 \pm 5.61\%$ | $49.85 \pm 2.65\%$ | $71.69 \pm 0.64\%$ | $75.17 \pm 0.39\%$ | $78.01 \pm 0.60\%$ |
| G-NOISYSIGN-VOTE W.O. PRIVACY AMPLIFICATION | $38.00 \pm 5.26\%$ | $49.85 \pm 3.22\%$ | $71.64 \pm 0.83\%$ | $75.84 \pm 0.55\%$ | $78.52 \pm 0.39\%$ |
| L-NOISYSIGN | $42.52 \pm 3.11\%$ | $58.39 \pm 1.53\%$ | $73.89 \pm 0.78\%$ | $77.18 \pm 0.31\%$ | $79.66 \pm 0.45\%$ |
| L-NOISYSIGN-VOTE | $40.46 \pm 3.82\%$ | $53.89 \pm 1.82\%$ | $73.23 \pm 0.47\%$ | $76.49 \pm 0.31\%$ | $79.24 \pm 0.31\%$ |

*Table 4.* Fashion-MNIST Test Accuracy for Varying Privacy Requirements Per Communication Round ($\alpha = 100$)

| $\mu$ | 0.04 | 0.08 | 0.4 | 0.8 | 1.6 |
|---|---|---|---|---|---|
| GAUSSIAN NOISE | $45.47 \pm 3.25\%$ | $59.21 \pm 2.25\%$ | $74.29 \pm 0.42\%$ | $77.57 \pm 0.23\%$ | $80.31 \pm 0.27\%$ |
| G-NOISYSIGN | $45.46 \pm 3.08\%$ | $59.84 \pm 2.40\%$ | $74.64 \pm 0.34\%$ | $77.61 \pm 0.44\%$ | $80.24 \pm 0.22\%$ |
| G-NOISYSIGN-VOTE | $42.42 \pm 3.40\%$ | $57.32 \pm 3.42\%$ | $73.60 \pm 0.29\%$ | $77.18 \pm 0.30\%$ | $79.81 \pm 0.28\%$ |
| L-NOISYSIGN | $45.26 \pm 3.19\%$ | $59.43 \pm 2.51\%$ | $74.91 \pm 0.55\%$ | $77.69 \pm 0.55\%$ | $80.28 \pm 0.23\%$ |
| L-NOISYSIGN-VOTE | $42.80 \pm 2.77\%$ | $55.42 \pm 2.56\%$ | $73.32 \pm 0.72\%$ | $77.00 \pm 0.32\%$ | $79.80 \pm 0.22\%$ |

*Table 5.* CIFAR-10 Test Accuracy for Varying Privacy Requirements Per Communication Round ($\alpha = 100$)

| $\mu$ | 0.8 | 1.6 | 4 | 8 | 16 |
|---|---|---|---|---|---|
| GAUSSIAN NOISE | $41.24 \pm 1.21\%$ | $48.60 \pm 0.53\%$ | $56.44 \pm 1.12\%$ | $63.63 \pm 0.79\%$ | $66.35 \pm 1.12\%$ |
| G-NOISYSIGN | $40.18 \pm 0.89\%$ | $47.90 \pm 0.64\%$ | $57.24 \pm 1.01\%$ | $63.18 \pm 1.27\%$ | $67.23 \pm 1.38\%$ |
| G-NOISYSIGN-VOTE | $38.98 \pm 0.75\%$ | $46.50 \pm 0.67\%$ | $55.60 \pm 0.68\%$ | $61.87 \pm 0.91\%$ | $66.16 \pm 1.00\%$ |
| L-NOISYSIGN | $40.39 \pm 0.97\%$ | $48.28 \pm 0.66\%$ | $57.73 \pm 0.93\%$ | $62.45 \pm 1.13\%$ | $66.82 \pm 1.71\%$ |
| L-NOISYSIGN-VOTE | $38.88 \pm 0.74\%$ | $46.76 \pm 1.02\%$ | $55.74 \pm 1.03\%$ | $61.65 \pm 1.20\%$ | $65.93 \pm 0.82\%$ |

*Table 6.* Fashion-MNIST Test Accuracy for Varying Privacy Requirements with 5 workers selected Per Communication Round ($\alpha = 0.1$)

| $\mu$ | 0.04 | 0.08 | 0.4 | 0.8 | 1.6 |
|---|---|---|---|---|---|
| GAUSSIAN NOISE | $27.90 \pm 3.69\%$ | $43.00 \pm 3.63\%$ | $69.06 \pm 0.98\%$ | $73.15 \pm 0.60\%$ | $76.06 \pm 0.97\%$ |
| G-NOISYSIGN | $28.53 \pm 4.42\%$ | $42.67 \pm 3.23\%$ | $68.93 \pm 1.33\%$ | $73.26 \pm 0.87\%$ | $76.37 \pm 0.59\%$ |
| G-NOISYSIGN-VOTE | $24.39 \pm 3.57\%$ | $37.81 \pm 2.08\%$ | $67.20 \pm 1.02\%$ | $72.33 \pm 0.78\%$ | $75.60 \pm 0.85\%$ |
| L-NOISYSIGN | $25.86 \pm 4.71\%$ | $42.44 \pm 2.95\%$ | $68.78 \pm 1.13\%$ | $73.18 \pm 0.96\%$ | $76.31 \pm 0.70\%$ |
| L-NOISYSIGN-VOTE | $24.17 \pm 4.75\%$ | $37.46 \pm 4.09\%$ | $66.92 \pm 1.25\%$ | $72.34 \pm 1.01\%$ | $75.69 \pm 0.55\%$ |

