# OpenReview forum: "Noisy SIGNSGD Is More Differentially Private Than You (Might) Think"
_ICML.cc/2025/Conference — ICML 2025 poster_

### Official Review · Reviewer_nyXX · 2025-02-25

**Overall Recommendation:** 4

**Summary:**

The authors study the privacy benefits of the map $sign(x)$ when combined with additive Gaussian noise in the Noisy SIGNSGD algorithm. They show that since $sign(x)$ drops the magnitude information, it indeed “amplifies” the privacy. The results show that the use of logistic noise may not be superior to using $sign(x)$ with additive Gaussian noise as reported in prior works when such a privacy amplification effect is considered. The claim is supported by both theoretical analysis and experiments on image classification tasks.

## update after rebuttal

I thank the authors for their explanation. I have no further questions, and I will keep my rating. I do think adding the comparison and discussion in the related works will further improve the manuscript.

**Claims And Evidence:**

Yes. I find the main claim is well supported.

**Essential References Not Discussed:**

I wonder how the Noisy SignSGD  (i.e., 1-bit compression) is compared to the sketching-based approaches, such as [1] and the relevant literature? I feel it is interesting to also see the trade-off along the communication complexity aspect in greater detail. Nevertheless, I think it is also fine to just focus on the Noisy SignSGD-based method. This question is just out of curiosity.

### References
[1] Optimal compression of locally differentially private mechanisms, Shah et al. AISTATS 2022.

**Experimental Designs Or Analyses:**

The experiment results show that using Noisy SignSGD can give comparable utility while saving 32x communication overhead since only one bit is transmitted.

**Methods And Evaluation Criteria:**

Yes. The main claim on the privacy benefits of $sign()$ is well supported by not only theoretical analysis but also numerical results to illustrate the resulting privacy bound compared to the one leveraging logistic noise.

**Other Comments Or Suggestions:**

No.

**Other Strengths And Weaknesses:**

The paper is well-written, and the problem is very well-motivated. While the analysis is not super technical, I like the fact that the authors summarized their ideas in a way that the general privacy audience can understand. It is a joyful read for me. I do not find any major weaknesses in the work.

**Questions For Authors:**

Maybe the question is out of the scope, but I wonder how Noisy SignSGD, especially when adopting the improved privacy analysis from the authors, compared to the sketching-based approaches? What is the corresponding privacy-utility-communication trade-off? It is totally fine if the authors cannot answer this question for now, but I feel this is a very interesting question to think about for distributed learning with DP constraints in general.

**Relation To Broader Scientific Literature:**

The finding is quite generic. Given that 1-bit quantization (i.e., signSGD) is indeed important in some settings, I believe the result of the paper can benefit the community.

**Theoretical Claims:**

I did not check the proof in the appendix but the proof sketch provided in the main text makes sense to me.

---

> ### Author Rebuttal · Authors · 2025-03-31
>
> Dear reviewer nyXX
>
> We appreciate your time and effort in reviewing our paper and providing a positive evaluation. Please find our response below.
>
> **Comparison with sketching-based approaches**: Thank you for pointing out this important aspect. We note that the compression of differentially private mechanisms aims to compress and simulate the distribution of the DP randomizer, usually in the presence of some shared randomness. The resulting compressed mechanisms have a smaller communication cost compared to the original mechanism while retaining the (or weakened) privacy guarantee.
>
> On one hand, our work endeavors to quantify the privacy amplification effect of the sign-based compressor instead of achieving minimum communication cost for a given differential privacy mechanism without ruining its privacy guarantee. On the other hand, we believe that our method can be further combined with these compression schemes (i.e., simulating a Bernoulli distribution). Considering the privacy-utility-communication trade-off in this case would be an interesting future direction.
>
> We will add a literature review and the corresponding discussions in related works in the revised version.

---

> > ### Comment · Reviewer_nyXX · 2025-04-04
> >
> > I thank the authors for their explanation. I have no further questions, and I will keep my rating. I do think adding the comparison and discussion in the related works will further improve the manuscript.

---

### Official Review · Reviewer_t9hH · 2025-03-12

**Overall Recommendation:** 4

**Summary:**

This paper investigates how sign-based gradient compression–specifically, Noisy signSGD–can inherently amplify differential privacy.
In a distributed learning scenario, the authors introduce theoretical analysis for the privacy guarantees under the f-DP framework and compare to two variants: G-NoisySign and L-NoisySign.

**Claims And Evidence:**

Privacy amplification via compression: The paper rigorously shows that the act of discarding the gradient magnitudes via the sign operator amplifies differential privacy. In the theoretical analysis, the authors derive tight privacy bounds that quantify this effect.

**Essential References Not Discussed:**

The key contribution of this work is to propose f-DP-based federated learning. Since the idea of the f-DP with federated learning is also discussed in the paper below, the reviewer recommends to cite one additional paper.

```
@inproceedings{zheng2021federated,
  title={Federated f-differential privacy},
  author={Zheng, Qinqing and Chen, Shuxiao and Long, Qi and Su, Weijie},
  booktitle={International conference on artificial intelligence and statistics},
  pages={2251--2259},
  year={2021},
  organization={PMLR}
}
```

**Experimental Designs Or Analyses:**

In the experiments, the authors show that the proposed approach achieves the privacy-utility trade-off comparable to the classical DP-SGD. The experimental design makes sound.

**Methods And Evaluation Criteria:**

The methods and the experimental results are appropriate to the problem at hand. However, the number of repeats (5) is small for generalization.

**Other Comments Or Suggestions:**

- I think the analysis with the linear mean estimation (Eqs. (14)-(16)) is not an appropriate method because it is not optimal. For example, if we have $E[f(x)] = x + 0.1 x^2$, $x\in[0,1]$, we may not treat the $x^2$ term as an error. If we use a proper linear estimator, the relative magnitude of the second term in the mean is not an error.
- In Figure 3, like dithering, a proper amount of the additive noise is helpful for successful aggregation. In that situation, the aggregation is always wrong if $\sigma_{DP}$ is near zero. Inversely, it seems like G-NoisySign offers larger noise std compared to L-NoisySign if $\sigma_{DP}$ is small. Can you please add a discussion for this result?
- Also, regarding Figure 3, the reviewer suggests adding figures for more general cases. Ex)10 workers are selected for each round, $x_m=-1.0 + \mathcal{N}(0,1)$ for $m<7$ and $x_m= -1.0 + \mathcal{N}(0,1)$ o.w.
- The reviewer suggests the authors add **more repeats** for the experimental results.
- The authors mentioned that the G-NoisySign is more suitable for distributed learning with heterogeneous data. The reviewer suggests adding experimental results for **the homogeneous dataset (i.i.d.) setting** for verification, like Tables 1 and 2.
- The reviewer suggests adding more results with a smaller number of participating workers. For example, 5 workers are participating in each communication found, like Tables 1 and 2.
- (Optional) Consider deeper neural networks with more complex datasets (e.g., cifar100).

**Other Strengths And Weaknesses:**

**Strengths**: This paper provides a tight bound based on the neyman-pearson Lemma. Due to the tight bound, the proposed method in the experimental results achieves (or outperforms) the DP-SGD, whereas previous studies have a slightly lower accuracy compared to the DP-SGD.

**Weakness**: Despite their theoretical contributions, the performance gap between the proposed method and the baselines are marginal/

**Questions For Authors:**

- In the numerical results (e.g., table 1), they said that they run all the algorithms for 5 repeats and **present the best results**. What is the correct one.. However, it seems like **mean accuracy** is reported. The reviewer guesses they run 5 repeats and present the mean results and the error bar (std) of the results. Is it really the best results with 5 repeats?

**Relation To Broader Scientific Literature:**

There have been many papers for DPSGD and signSGD. The reviewer thinks that this paper provides a very tight bound for the DP + signSGD.

**Theoretical Claims:**

All the theoretical analyses are well written; however, the reviewer raises an issue in the error analysis (lines 233-), the comparison of the **estimation error**. For more details, please refer to the additional comment section.

---

> ### Author Rebuttal · Authors · 2025-03-31
>
> Dear reviewer t9hH
>
> We appreciate your time and effort in reviewing our paper and providing constructive comments. Please find our point-by-point response below.
>
> **Question about error analysis:** The rationale behind the analysis in Eqs (14)-(16) is that unbiased estimate of gradients is generally preferred in distributed/federated learning, considering that the majority of convergence analyses are established on the assumption of unbiased gradient estimator. Particularly, assuming a smooth loss function, the key is to bound $\langle\nabla F(\boldsymbol{w}^{(t)}), \boldsymbol{w}^{(t+1)}-\boldsymbol{w}^{(t)}\rangle$ (see Eq. (71)), where $\boldsymbol{w}^{(t+1)}-\boldsymbol{w}^{(t)} = -\eta\frac{1}{M}\sum_{i\in \mathcal{H}}\boldsymbol{g}\_{i}\^{(t)}$ for SGD and $-\eta\frac{1}{M}\sum\_{i\in \mathcal{H}}sign(\boldsymbol{g}\_{i}\^{(t)} + \boldsymbol{n}\_{i}\^{(t)})$ for Noisy SignSGD. In this case, it is desired that $\mathbb{E}[sign(\boldsymbol{g}\_{i}\^{(t)} + \boldsymbol{n}\_{i}\^{(t)})]$ is as close to (a scaled version of) $\boldsymbol{g}_{i}^{(t)}$ as possible such that the performance of Noisy SignSGD approaches that of SGD.
>
> **Results in Figure 3:** Indeed, for small $\sigma_{DP}$, Gaussian noise $\mathcal{N}(0,\sigma_{DP})$ tends to have a larger standard deviation than Logistic noise $Logistic(0,s)$. More specifically, to ensure the same privacy guarantees, $s = c/\ln(\Phi(c/\sigma_{DP})/\Phi(-c/\sigma_{DP}))$ (see discussion below Eq.(13)), and $\sigma_{DP} > s\pi/\sqrt{3}$ (the standard deviation of logistic noise) when $\sigma_{DP}$ is small. In this sense, G-NoisySign may not always be better than L-NoisySign, especially when $\sigma_{DP}$ is small and data heterogeneity is not a concern. However, we note that heterogeneous data and a large $\sigma_{DP}$ (i.e., more stringent privacy requirement) are of particular interest, especially in differentially private federated learning. We will add the discussion and numerical results for more general cases in the revised version.
>
> **Additional experiments**: We perform additional experiments on FMNIST with $\alpha = 100$ to simulate homogeneous data, and the results (10 repeats) are given below.
>
> |$\mu$|0.04|0.08|0.4|0.8|1.6|
> |--------|-------|-------|-------|-------|-------|
> |Gaussian Noise|$45.47\pm3.25\\%$|$59.21\pm2.25\\%$|$74.29\pm 0.42\\%$|$77.57\pm0.23\\%$|$80.31\pm 0.27\\%$|
> |G-NoisySign|$45.46\pm3.08\\%$|$59.84\pm2.40\\%$|$74.64\pm0.34\\%$|$77.61\pm0.44\\%$|$80.24\pm0.22\\%$
> |G-NoisySign-Vote|$42.42\pm3.40\\%$|$57.32\pm3.42\\%$|$73.60\pm0.29\\%$|$77.18\pm0.30\\%$|$79.81\pm 0.28\\%$
> |L-NoisySign|$45.26\pm3.19\\%$|$59.43\pm2.51\\%$|$74.91\pm0.55\\%$|$77.69\pm0.55\\%$|$80.28\pm0.23\\%$
> |L-NoisySign-Vote|$42.80\pm2.77\\%$|$55.42\pm2.56\\%$|$73.32\pm0.72\\%$|$77.00\pm0.32\\%$|$79.80\pm0.22\\%$
>
> We also perform more experiments on CIFAR-10 (5 repeats due to limited time) with $\alpha = 100$.
>
> |$\mu$|0.8|1.6|4|8|1.6|
> |--------|-------|-------|-------|-------|-------|
> |Gaussian Noise|$41.31\pm1.11\\%$|$48.67\pm0.41\\%$|$56.65\pm0.92\\%$|$63.95\pm0.86\\%$|$67.07\pm1.30\\%$|
> |G-NoisySign|$40.47\pm0.78\\%$|$48.13\pm0.78\\%$|$57.48\pm1.03\\%$|$63.41\pm0.45\\%$|$67.21\pm0.50\\%$
> |G-NoisySign-Vote |$38.91\pm0.44\\%$|$46.75\pm0.59\\%$|$55.87\pm0.67\\%$|$61.67\pm1.09\\%$|$66.38\pm0.69\\%$
> |L-NoisySign|$40.85\pm0.33\\%$|$48.55\pm0.50\\%$|$57.32\pm0.91\\%$|$62.63\pm1.15\\%$|$67.79\pm0.75\\%$
> |L-NoisySign-Vote|$39.08\pm0.68\\%$|$46.45\pm1.07\\%$|$55.10\pm0.90\\%$|$62.06\pm0.53\\%$|$65.70\pm1.00\\%$
>
> **Fewer workers**: We perform more experiments on FMNIST (10 repeats) with 5 workers selected in each round with $\alpha = 0.1$.
> |$\mu$|0.04|0.08|0.4|0.8|1.6|
> |--------|-------|-------|-------|-------|-------|
> |Gaussian Noise|$27.90\pm3.69\\%$|$43.00\pm3.63\\%$|$69.06\pm0.98\\%$|$73.15\pm0.60\\%$|$76.06\pm0.97\\%$|
> |G-NoisySign|$28.53\pm4.42\\%$|$42.67\pm3.23\\%$|$68.93\pm1.33\\%$|$73.26\pm0.87\\%$|$76.37\pm0.59\\%$
> |G-NoisySign-Vote| $24.39\pm3.57\\%$ |$37.81\pm2.08\\%$|$67.20\pm1.02\\%$|$72.33\pm0.78\\%$|$75.60\pm0.85\\%$
> |L-NoisySign|$25.86\pm4.71\\%$|$42.44\pm2.95\\%$|$68.78\pm1.13\\%$|$73.18\pm0.96\\%$|$76.31\pm0.70\\%$
> |L-NoisySign-Vote|$24.17\pm4.75\\%$|$37.46\pm4.09\\%$|$66.92\pm1.25\\%$|$72.34\pm1.01\\%$|$75.69\pm0.55\\%$
>
> It can be observed that G-NoisySign and L-NoisySign attain comparable performance with the Gaussian mechanism in all the experiments above.
>
> **Performance gap with baselines & best results:** Note that the goal of our paper is not to outperform DP-SGD in accuracy but to show the privacy amplification of sign-based compression. Please refer to our response to Reviewer AkDc under **Marginal improvement over DP-SGD** and **Concern about presenting the best results** for more details.
>
> **Number of repeats & essential references:** We will increase the number of repeats and add the reference as suggested in our revised version.
>
> We hope that your comments have been addressed adequately. Please let us know if there are any questions.

---

> > ### Comment · Reviewer_t9hH · 2025-04-04
> >
> > ## Round 1
> > Sincerely sorry for my late response. Most of the concerns have been cleared.
> >
> > - **Question about error analysis:** Thank you for the authors' response.
> > - **Results in Figure 3:** I want to see the results of the more general cases I mentioned before. Can you please describe how Figure 3 changes in general cases? I guess the results will be flipped if the setting goes to homogeneous cases. I think this might help to strengthen a bit more what this paper is claiming and in what situations it might be beneficial.
> >   - Because the example provided in Figure 3 is an extreme case, it cannot represent the general case. Consider the minimum wrong aggregation probability is higher than 0.4.
> > - **Additional experiments:** Thank you for the authors' efforts. Everything is clear now. I believe the slight degradation of G-NoisySign compared to L-NoisySign is due to the homogeneous setting ($\alpha=100.0$).
> >
> > ---
> > ## Round 2
> >
> > Dear authors,
> >
> > I sincerely appreciate your kind response to my questions. I confirm that all the comments I raised are clearly addressed.
> >
> > - **Suggestion (optional and does not affect the score):** I have one additional comment about the assumption for the bounded gradient. For the FL scenarios, many studies use the gradient heterogeneity assumption instead of the bounded norm to consider heterogeneity of the data distribution. I think it will better represent how the non-i.i.d. datapoint distribution affects the convergence.
> >
> > Anyway, since my initial questions are clearly resolved, I have changed the recommendation from (3: weak accept) to (4:accept).
> > If you have any further questions regarding my suggestion, please let me know.
> >
> > Thank you for your efforts to respond to my questions.

---

> > > ### Author Response · Authors · 2025-04-05
> > >
> > > Dear Reviewer t9hH,
> > >
> > > We appreciate your further comments and questions, which greatly help improve our paper.
> > >
> > > --------
> > >
> > > **Results in Figure 3**: Indeed, for less heterogeneous cases, L-NoisySign could outperform G-NoisySign. To alleviate your concern, we further describe how Figure 3 changes here. We perform some additional experiments, in which a set of 10 workers are considered, with $x_{m} = \mathcal{N}(-1,1)$ for $m < 7$ and $x_{m} = \mathcal{N}(x_{heterog},1)$ otherwise. We examine
> > > $x_{heterog} \in \\{1, 3, 5, 10\\}$ for different levels of data heterogeneity. It is observed that when the data distribution is less heterogeneous (i.e., $x_{heterog} = 1$), L-NoisySign outperforms G-NoisySign, and the probability of wrong aggregation is always smaller than 0.5. In this case, the vanilla SIGNSGD converges well, and Gaussian noise tends to result in a larger variance than Logistic noise with the same privacy guarantees. When the data distribution becomes more heterogeneous (i.e., $x_{heterog} = 3$), G-NoisySign first outperforms L-NoisySign for an appropriately small $\sigma_{DP}$ and then underperforms when $\sigma_{DP}$ keeps increasing (i.e., there exists a crossover point). Intuitively, G-NoisySign yields a smaller probability of wrong aggregation when adding some appropriate noise helps (like dithering), and the performance might degrade when the noise is too large. As data heterogeneity becomes more severe, the crossover happens at a larger $\sigma_{DP}$ (and the difference in probability of wrong aggregation becomes negligible). Moreover, we observe that G-NoisySign attains a smaller minimum probability of wrong aggregation than L-NoisySign for $x_{heterog} \in \\{3, 5, 10\\}$. This validates that G-NoisySign may be more suitable for heterogeneous cases.
> > >
> > > It is worth mentioning that since $x_{m}$'s are Gaussian in this case, the level of data heterogeneity is somewhat reduced. Therefore, we also examine the scenarios where $x_{m}$'s are fixed, and the results agree with the discussion above.
> > >
> > > We provide the corresponding numerical results in the anonymous link https://anonymous.4open.science/r/AnonymousICML25Rebuttal-05F8/, and will revise the manuscript accordingly.
> > >
> > > --------
> > >
> > > Thank you again for reading the rebuttal. We hope that your comments and concerns have been adequately addressed, and if this is the case, it would be great if you are willing to kindly increase your score. Please let us know if there are any further questions.
> > >
> > > --------
> > >
> > > ## Round 2
> > >
> > > Dear Reviewer t9hH,
> > >
> > > We appreciate your further comments and kind suggestions.
> > >
> > > **Regarding the bounded gradient assumption**, we agree that using the gradient dissimilarity assumption will provide a more accurate characterization of the impact of data heterogeneity. In fact, the bounded gradient assumption implies the bounded gradient dissimilarity assumption (and, therefore, is stronger). The bounded gradient assumption in this work is not introduced to alleviate the difficulty in convergence analyses caused by data heterogeneity but the bias introduced by gradient clipping. Several existing works [R1, R2] have considered the highly non-trivial impact of clipping on DP-SGD. The joint impact of sign-based compression and gradient clipping is more complicated and challenging to analyze without the bounded gradient assumption, which is an interesting direction for future work.
> > >
> > > Thanks again for your constructive comments and suggestions.
> > >
> > > -------
> > >
> > > [R1] X. Zhang, et al. Understanding clipping for federated learning: Convergence and client-level differential privacy, ICML 2022.
> > >
> > > [R2] X. Chen, et al. Understanding gradient clipping in private SGD: A geometric perspective, NeurIPS, 2020

---

### Official Review · Reviewer_Jupt · 2025-03-13

**Overall Recommendation:** 4

**Summary:**

SignSGD is a technique to compress a gradient in order to reduce its communication cost. It is typically applied in decentralized or federated SGD where the transmission of gradients is a regular operation. The key idea behind it is to transmit the sign of each component, reducing the communication cost to one bit per dimension.

The paper studies the differential privacy (DP) guarantees of SignSGD. Current techniques to make SignSGD differentially private rely on the addition of noise before compression. They focus on making the uncompressed gradient differentially private which, by post-processing, implies the differential privacy guarantees of the compressed gradients. However, these techniques ignore  the privacy amplification obtained by the sign compression. The current contribution quantifies this amplification, showing that less noise is required to obtain DP, which results in more accurate models.

Under the framework of $f$-DP, the paper shows that Noisy SignSGD using Gaussian or Logistic noise achieves similar performance than the uncompressed version despite the 32x improvement factor in compression.

## update after rebuttal
Authors have clarified my concerns. Therefore I keep my score supporting acceptance.

**Claims And Evidence:**

Appropriate theory and experiments back the claims of the paper.

**Essential References Not Discussed:**

To the best of my knowledge, there are no essential references missed in the paper.

**Experimental Designs Or Analyses:**

The experiment design is reasonable. However, it would have been more informative to show the convergence across iterations to have an impression of the convergence speed of the evaluated techniques instead of just the final accuracy in Table 1.

**Methods And Evaluation Criteria:**

The evaluation setting is reasonable for the current problem.

**Other Comments Or Suggestions:**

I have only minor comments:
- The third sentence of the abstract is a bit long and complicated. It could be reformulated for clarity
- The font size of Figure 2 is too small and should be increased

**Other Strengths And Weaknesses:**

I have enjoyed reading the paper. It is in general well written and provides clear explanations of each key concept.

**Questions For Authors:**

Please address the broader positioning to other privacy preserving compressors mentioned in the review.

**Relation To Broader Scientific Literature:**

The paper seems to reasonably address its relation with respect to sign based compressors. However, it does not position itself with respect to other compression techniques under privacy constraints such as [R1-R5] listed below.

References:

[R1] Triastcyn, Aleksei, Matthias Reisser, and Christos Louizos. "Dp-rec: Private & communication-efficient federated learning." arXiv preprint arXiv:2111.05454 (2021).
[R2] Bassily, Raef, and Adam Smith. "Local, private, efficient protocols for succinct histograms." Proceedings of the forty-seventh annual ACM symposium on Theory of computing. 2015.
[R3] Feldman, Vitaly, and Kunal Talwar. "Lossless compression of efficient private local randomizers." International Conference on Machine Learning. PMLR, 2021.
[R4] Shah, Abhin, et al. "Optimal compression of locally differentially private mechanisms." International Conference on Artificial Intelligence and Statistics. PMLR, 2022.
[R5] Liu, Yanxiao, et al. "Universal exact compression of differentially private mechanisms." Advances in Neural Information Processing Systems 37 (2024): 91492-91531

**Theoretical Claims:**

I have checked the proofs of Theorems 1 and 2, which seem correct.

---

> ### Author Rebuttal · Authors · 2025-03-31
>
> Dear Reviewer Jupt,
>
> We appreciate your time and effort in reviewing our paper and providing a positive evaluation. Please find our point-by-point response below.
>
> **Relation to other compression techniques under privacy constraints:** Thank you for pointing out this important aspect. We note that the compression of differentially private mechanisms aims to compress and simulate the distribution of the DP randomizer, usually in the presence of some shared randomness. The resulting compressed mechanisms have a smaller communication cost compared to the original mechanism while retaining the (or weakened) privacy guarantee.
>
> On one hand, our work endeavors to quantify the privacy amplification effect of the sign-based compressor instead of achieving minimum communication cost for a given differential privacy mechanism. On the other hand, we believe that our method can be further combined with these compression schemes. The problem concerning privacy-utility-communication trade-off is important and interesting, which will be considered in our future work.
>
> We will add a literature review and the corresponding discussions in related works in the revised version.
>
> **Convergence across iterations:**  We will add the corresponding figures showing the convergence with respect to the communication round or communication overhead in the revised version.
>
> **Third sentence of the abstract:** We will revise and break it down into two sentences for clarity.
>
> **The font size of Figure 2:** We will revise accordingly.
>
> We hope that your comments have been addressed adequately. Please let us know if there are any questions.

---

> > ### Comment · Reviewer_Jupt · 2025-04-07
> >
> > Dear Authors,
> >
> > Thank you for your reply. I have no further comments or concerns. I will confirm my score after the discussion with other reviewers.

---

### Official Review · Reviewer_AkDc · 2025-03-19

**Overall Recommendation:** 2

**Summary:**

This paper considers the privacy guarantees of Noisy SignSGD, an algorithm that adds noise to a value, then releases its sign.
It is shown that releasing the sign of the value, rather than the noisy value itself, amplifies the privacy guarantees.
A method of majority vote for aggregating the sign gradient is also proposed.
Numerical experiments confirm that the proposed method, despite its biased nature, is on-par with the classical DP-SGD algorithm.

**Claims And Evidence:**

Theoretical claims are provided with full proofs, and empirical claims are in line with the empirical results.

**Essential References Not Discussed:**

Not to my knowledge.

**Experimental Designs Or Analyses:**

Experimental design is rather sound, with reasonable choice of datasets and problems and a wide range of hyperparameter choice.

However, clipping threshold is set arbitrarily to $C=1$ or $C=2$, which may have an important impact on the behaviour of the algorithm, especially since it relies only on the sign.

The statement "We run all the algorithms for 5 repeats and present the best results" is slightly concerning, as it is a bit unusual in my opinion.
Generally, one would rather report the average and standard deviation rather than maximum and standard deviation.

**Methods And Evaluation Criteria:**

Benchmark datasets and the used baselines make sense.

**Other Comments Or Suggestions:**

It seems that sign SGD could allow gradients not to be clipped before applying the sign function.
For instance, one may add noise to the gradient (possibly using the gradient's scale). This does not provide privacy in itself, but this could be enforced later using mechanisms like randomized response to perturb the sign of the gradient.
The resulting procedure could allow to give DP guarantees without requiring to know the scale of the gradient before-hand.
As such, no clipping is required when computing the gradient, while privacy is preserved by randomized response.

I understand this is a complicated question, and mention this as a potential direction for future research, but would it be the case that randomizing the sign of the algorithm could further improve the privacy guarantees and help to have additional amplification?

**Other Strengths And Weaknesses:**

**Strengths**
1. The paper shows that Sign SGD can enhance privacy guarantees, which is in line with the intuition that releasing only the sign can improve privacy guarantees as less information is released.
2. Numerical experiments show that the proposed methods is on par with DP-SGD for a given privacy guarantee.

**Weaknesses.**
1. While it is shown that privacy guarantees are amplified, the amplification is rather small. This should rather be referred to as a "tighter evaluation of the privacy of sign SGD" rather than proper amplification. This is in line with Figure 1, which shows that the analysis is tighter rather than shows amplification (e.g. with noise to add divided by the magnitude of the gradient).
2. Majority voting as an aggregation procedure is interesting, but almost systematically decrease accuracy in experiments, which is a bit disappointing.
3. Theoretical guarantees for noisy sign SGD are rather weak, and do not account for clipping, although one could expect that clipping has a somewhat important impact on the result.
4. Experiments only highlight marginal improvement from sign SGD in comparison with DP-SGD with Gaussian noise. Moreover, step size is heavily tuned: in sign method, this may lead to "guessing the scale of the gradient by hyperparameter tuning". Tuning this hyperparameter may have less impact on DP-SGD.

**Questions For Authors:**

1. It seems that "amplification" is rather small: are there settings where one can obtain arbitrarily large gains, or is amplification bounded by a constant factor? (In which case I would rather talk about "tighter analysis of DP guarantees of noisy SignSGD".)
2. How important is the hyperparameter tuning step for noisy Sign SGD, in comparison with DP-SGD? In particular, is one of the two algorithms more sensitive to hyperparameter tuning than the other?

**Relation To Broader Scientific Literature:**

Related work with respect to sign SGD and to differential privacy are discussed.

**Theoretical Claims:**

Not in all details, but derivations presented in the main text seem correct.

---

> ### Author Rebuttal · Authors · 2025-03-31
>
> Dear Reviewer AkDc,
>
> We appreciate your time and effort in reviewing our paper and providing constructive comments. Please find the response to the comments below.
>
> **Concern about presenting the best results:** This is due to confusion. We run the algorithms for 5 repeats, compute the mean and standard deviation, and present the results with the highest mean accuracy over all the tuned learning rates. We will revise it accordingly to avoid confusion.
>
> **Privacy amplification or tighter evaluation:** We note that in Fig. 1, the black curve shows the privacy of the Gaussian mechanism, and the blue curve for the Gaussian mechanism combined with $sign$. The improvement is due to the privacy amplification effect of $sign$.
>
> **Privacy amplification is small:** As we discussed in Remark 2, compared to the Gaussian mechanism without $sign$, there is an improvement by a factor of $\sqrt{\frac{\pi}{2}} \approx 1.25$. In such a case, if a Gaussian mechanism with variance $\sigma$ gives $\mu_{G} = \frac{2C}{\sigma}$-GDP, further incorporating $sign$ provides $\mu_{GNS} = \frac{2C}{\sigma\sqrt{\frac{\pi}{2}}} \approx 0.8\mu_{G}$. There is a decrease in $\mu$ for around 20\%.
>
> **Accuracy decrease for majority vote:** Note that majority vote improves the server-to-client communication efficiency and may provide another improvement in privacy if central DP is concerned (consider privacy leakage of releasing the aggregated results), considering that taking the majority vote is equivalent to adopting $sign$ at the server side. Unfortunately, the improvement in communication efficiency and central DP comes with a loss in privacy. The interlay among communication, privacy in terms of central DP, and accuracy would be an interesting future direction.
>
> **Impact of clipping:** We agree that studying the impact of clipping is important, which has been considered in some existing works, e.g., [R1-R2]. However, we want to emphasize that the convergence analyses are not our major contributions. Instead, we focus more on the privacy amplification effect of sign-based compressors. Therefore, we follow the existing literature and adopt the bounded gradient assumption.
>
> [R1] X. Zhang, et al. Understanding clipping for federated learning: Convergence and client-level differential
> privacy, ICML 2022.
>
> [R2] X. Chen, et al. Understanding gradient clipping in private SGD: A geometric perspective, NeurIPS, 2020
>
> **Marginal improvement over DP-SGD:** We would like to clarify that our goal is not to show that noisy SignSGD outperforms DP-SGD in accuracy, but that compression by $sign$ does not result in much loss in privacy-accuracy tradeoff. Particularly, discarding the magnitude leads to a significant improvement in communication efficiency (32$\times$ assuming each float number is represented by 32 bits), which may in turn hinder the accuracy. We show that the loss in accuracy caused by compression leads to privacy amplification. A comparable privacy-accuracy tradeoff with DP-SGD means that the improvement in communication efficiency is obtained almost for free.
>
> **Randoming the sign:** Randomizing the sign could further improve the privacy guarantees, and Theorem 1 in the paper can capture the amplification as long as the probability of flipping is known. However, without clipping, we may not utilize the privacy amplification effect of taking the signs since it is not DP on its own.
>
> **Step size tuning**: In our experiments, we find that DP-SGD is more sensitive to learning rates. The best learning rates of DP-SGD vary for different $\mu$'s, while those of G-NoisySign and L-NoisySign (and the majority vote variants) only change slightly. This may be attributed to the fact that adding noise with a large variance to the gradients makes the training process less stable (e.g., a large noise may lead to a large deviation in model updates), and a smaller learning rate is preferred. The clipping effect of the $sign$ compressor may help in this case. We will add more discussion in the revised version.
>
> **The choice of clipping threshold:** We note that many existing works adopt a similar clipping threshold for experiments at a similar scale. For example, [R3] suggested a consistent performance with $C \in \\{1,2,3,4,5\\}$ for MNIST. [R4] adopted $C=1$ for both MNIST and CIFAR-10. [R5] set $C \in \\{1, 1.5\\}$ for MNIST. In our experiments, we tested different $C \in \\{1,2\\}$ for FMNIST, and the trends are consistent with those presented in the paper. We will add more detailed discussions in the revised version.
>
> [R3] M. Abadi, et al. Deep learning with differential privacy. ACM SIGSAC conference on computer and communications security, 2016.
>
> [R4] Q. Zheng, et al. Federated f-differential privacy. AISTATS, 2021
>
> [R5] J. Dong, et al. Gaussian differential privacy. Journal of the Royal Statistical Society: Series B, 2022
>
> We hope that your comments have been addressed adequately. Please let us know if there are any questions.

---

> > ### Comment · Reviewer_AkDc · 2025-04-07
> >
> > Thank you for your answer. I understand that this can be seen as some kind of "amplification", although this is a very small amplification (about 1.25x), the phenomenon is interesting to study, and the result is nice.
> >
> > Nonetheless, I still think that the title "Noisy SIGNSGD Is More Differentially Private Than You (Might) Think" is a bit of an overstatement, since we are talking about a 1.25x amplification.
> >
> > > "We would like to clarify that our goal is not to show that noisy SignSGD outperforms DP-SGD in accuracy, but that compression by $sign$ does not result in much loss in privacy-accuracy tradeoff."
> >
> > I understand, and I agree this is an interesting phenomenon, especially in settings where communication bandwidth is limited. Nonetheless, **this phenomenon mostly seems due to the fact that variance dominates the bias in such problems, rather than to the amplification of privacy by a factor at most 1.25**. This can not be seen in the current experiments, that only include the "amplified noisy-signSGD" and the classical DP-SGD with Gaussian noise.
> >
> > > "However, without clipping, we may not utilize the privacy amplification effect of taking the signs since it is not DP on its own."
> >
> > It could be if randomizing the sign, and this could have more impact on "amplification".

---

> > > ### Author Response · Authors · 2025-04-08
> > >
> > > Dear Reviewer AkDc,
> > >
> > > We sincerely appreciate your further comments and questions.
> > >
> > > -------------
> > >
> > > The composition property in Lemma 2 suggests that, for the same overall privacy guarantee, **utilizing the privacy amplification effect allows for $\frac{\pi}{2} \approx 1.5\times$ training steps with the same Gaussian noise variance per step**. We believe that 50% more training steps is meaningful in improving the overall performance, as also suggested in [R6]. To alleviate your concern, due to limited time, we further perform some additional experiments on Fashion-MNIST ($\alpha = 0.1$, 10 repeats). Particularly, we add two more baselines: (1) G-NoisySign without utilizing the privacy amplification analyses (i.e., adding Gaussian noise with the same variance as DP-SGD); (2) G-NoisySign that runs 320 rounds (slightly more than $500*2/\pi$ rounds) instead of 500 rounds. The results are given below. **It can be observed that G-NoisySign and G-NoisySign-Vote outperform the two additional baselines in all the examined privacy budgets, which validates that the privacy amplification indeed leads to improvement in test accuracy.** Note that the results for the 5 algorithms in the manuscript are slightly different since we run more repeats here.
> > >
> > > |$\mu$|0.04|0.08|0.4|0.8|1.6|
> > > |--------|-------|-------|-------|-------|-------|
> > > |Gaussian Noise|$43.78\pm1.98\\%$|$58.69\pm1.48\\%$|$73.48\pm 0.49\\%$|$77.17\pm0.51\\%$|$79.90\pm 0.25\\%$|
> > > |G-NoisySign|$45.24\pm2.51\\%$|$58.46\pm2.48\\%$|$73.78\pm0.57\\%$|$77.20\pm0.29\\%$|$79.57\pm0.57\\%$
> > > |G-NoisySign (320 Round)|$40.94\pm4.49\\%$|$54.49\pm2.68\\%$|$72.71\pm0.68\\%$|$76.15\pm0.46\\%$|$78.70\pm0.65\\%$
> > > |G-NoisySign w.o. privacy amplification|$37.36\pm4.07\\%$|$52.06\pm2.54\\%$|$72.83\pm0.62\\%$|$76.49\pm0.43\\%$|$79.11\pm0.45\\%$
> > > |G-NoisySign-Vote|$42.18\pm4.03\\%$|$56.59\pm1.91\\%$|$73.23\pm0.66\\%$|$76.47\pm0.61\\%$|$79.27\pm 0.37\\%$
> > > |G-NoisySign-Vote (320 Round)|$35.79\pm5.61\\%$|$49.85\pm2.65\\%$|$71.69\pm0.64\\%$|$75.17\pm0.39\\%$|$78.01\pm0.60\\%$
> > > |G-NoisySign-Vote w.o. privacy amplification|$38.00\pm5.26\\%$|$49.85\pm3.22\\%$|$71.64\pm0.83\\%$|$75.84\pm0.55\\%$|$78.52\pm0.39\\%$
> > > |L-NoisySign|$42.52\pm3.11\\%$|$58.39\pm1.53\\%$|$73.89\pm0.78\\%$|$77.18\pm0.31\\%$|$79.66\pm0.45\\%$
> > > |L-NoisySign-Vote|$40.46\pm3.82\\%$|$53.89\pm1.82\\%$|$73.23\pm0.47\\%$|$76.49\pm0.31\\%$|$79.24\pm0.31\\%$
> > >
> > > -------
> > >
> > > **Randomizing the sign**: Indeed, further randomizing the signs on top of the current mechanism will add another level of privacy amplification, and the gain could be arbitrarily large (flipping the sign with a probability of 0.5 yields perfect privacy).
> > >
> > > -------
> > >
> > > We hope that your comments have been addressed adequately. Please let us know if there are any questions.
> > >
> > > [R6] J. Jang, et al., Rethinking dp-sgd in discrete domain: Exploring logistic distribution in the realm of signSGD. ICML, 2024.

---

### Decision · Program_Chairs · 2025-05-01

**Decision:**

Accept (poster)

**Comment:**

This paper analyzes a noisy variant of SignSGD that adds Gaussian noise for differential privacy before applying sign compression. The authors show that the sign operator enables slightly stronger privacy guarantees than standard post-processing. Most reviewers found the setting and results interesting, though they requested clarifications on several points. The rebuttal addressed many of these concerns and included new experiments demonstrating utility gains from the improved analysis, directly responding to one of Reviewer AkDc’s key concerns.

Considering the reviews, rebuttal, and discussion, I find this a solid contribution and recommend acceptance.